# Autoencoders as Cross-Modal Teachers: Can Pretrained 2D Image Transformers Help 3D Representation Learning?

**Runpei Dong**[1]   **Zekun Qi**[1]   **Linfeng Zhang**[2]   **Junbo Zhang**[2]   **Jianjian Sun**[3]   **Zheng Ge**[3]
**Li Yi**[2, 4, 5][†]       **Kaisheng Ma**[2][†]
[1] Xi'an Jiaotong University   [2] Tsinghua University   [3] MEGVII Technology[‡]
[4] Shanghai Artificial Intelligence Laboratory   [5] Shanghai Qi Zhi Institute

## Abstract

The success of deep learning heavily relies on large-scale data with comprehensive labels, which is more expensive and time-consuming to fetch in 3D compared to 2D images or natural languages. This promotes the potential of utilizing models pretrained with data more than 3D as teachers for cross-modal knowledge transferring. In this paper, we revisit masked modeling in a unified fashion of knowledge distillation, and we show that foundational Transformers pretrained with 2D images or natural languages can help self-supervised 3D representation learning through training **A**utoencoders as **C**ross-Modal **T**eachers (**ACT**). The pretrained Transformers are transferred as cross-modal 3D teachers using discrete variational autoencoding self-supervision, during which the Transformers are frozen with prompt tuning for better knowledge inheritance. The latent features encoded by the 3D teachers are used as the target of masked point modeling, wherein the dark knowledge is distilled to the 3D Transformer students as foundational geometry understanding. Our ACT pretrained 3D learner achieves state-of-the-art generalization capacity across various downstream benchmarks, *e.g.*, 88.21% overall accuracy on ScanObjectNN. Codes have been released at `https://github.com/RunpeiDong/ACT`.

## 1 Introduction

In recent years, AI systems powered by data-driven deep learning have been deployed in various areas (LeCun et al., 2015; He et al., 2016; Vaswani et al., 2017). The advancements in computing hardware have largely facilitated machine intelligence developments, which also encourages an emerging paradigm of transferring models trained on broad data, *i.e.*, *foundational models* (Bommasani et al., 2021). Great success has been witnessed in natural language processing (NLP) (Devlin et al., 2019; Radford et al., 2018; 2019; Brown et al., 2020; Radford et al., 2021), where the models are designed to learn generic representations through self-supervised knowledge probing on data of extreme size. Since the rapid development of Transformer (Vaswani et al., 2017) in vision (Dosovitskiy et al., 2021; Liu et al., 2021b), various efforts have been made to spread this trend from NLP towards foundational 2D visual understanding (Bao et al., 2022; He et al., 2022b; Wang et al., 2022a).

Meanwhile, compared to 2D vision and NLP, this course towards foundational visual computing is significantly lagging in the 3D community. We ask: *What makes 3D representation learning more challenging than 2D vision or NLP?* We offer some analytical answers from the following three perspectives:

Table 1: Data pattern comparison.

| Format | Scale | Semantics |
|---|---|---|
| Language | Broad | Dense & Structured |
| RGB Pixel | Large | Sparse & Unstructured |
| Coordinates | Moderate | Sparse & Unstructured |

  i. **Architecture disunity.** Pioneering architectures like PointNet (Qi et al., 2017a;b) can only encode 3D coordinates and it is not applicable for *masked denoising autoencoding (DAE)* (Vincent et al., 2008; 2010; Devlin et al., 2019) which is proved successful in NLP and 2D vision (He et al., 2022b). *Transformers* (Vaswani et al., 2017) has now closed this architectural gap, which enables a unified representation across *all modality formats* (Wang et al., 2022a) and brings a great potential of extending DAE for 3D (Yu et al., 2022; Pang et al., 2022).

---

[†]Corresponding authors.

[‡]Work partially done during the internship of Runpei Dong (`runpei.dong@gmail.com`) at MEGVII.

ii. **Data desert.** In comparison to images and free-form languages, it is more difficult to collect and label 3D (Chang et al., 2015) or 4D (Liu et al., 2022b) data, which generally requires more expensive and labor-intensive efforts. In addition, 3D data are seriously lacking considering the *scale of data*[1]. This motivates the usage of *cross-modal knowledge transfer*. Recent works either jointly train with other modalities for more effective contrast (Afham et al., 2022) or directly fine-tune 2D Transformers pretrained on image data (Wang et al., 2022b).

iii. **Pattern difference.** Table 1 shows the data pattern comparison of languages, 2D images and 3D point clouds. It is observed that: (i) 3D point cloud is usually unstructured containing sparse semantics unlike the language. This leads to the discrete identification learning for BERT-style tokenizer (Devlin et al., 2019) on point clouds more difficult (Yu et al., 2022) (see Sec. 6.1). (ii) 2D images are regularly distributed on grids, while 3D point clouds irregularly sampled from the object surface. This structural difference leads to the difficulty of constructing contrastive targets both for single-modality augmentations (Hou et al., 2021) and for cross-modal correspondence (Li et al., 2022). (iii) How to design a better representation with enriched *semantics* becomes the *de-facto* principal for self-supervised 3D understanding.

Motivated by the analysis above, we propose to train **A**utoencoders as **C**ross-Modal **T**eachers (**ACT**). Our ACT utilizes foundational Transformers pretrained with 2D images or natural languages as cross-modal teachers, carrying profound knowledge and powerful representation capacity. In this way, the data desert issue in 3D is alleviated. Transformer is employed as the generic 3D learner, which closes the architectural gap toward masked modeling representation learning. By simply tuning pretrained Transformers as autoencoders on 3D data in a self-supervised fashion, the Transformers can consume and encode 3D point clouds into representations with rich semantics. In order to preserve and inherit the pretrained foundational knowledge, prompt tuning (Jia et al., 2022) is used during this procedure. As a result, our ACT makes the pretrained Transformers spontaneously cross-modal teachers that provide semantically enriched masked modeling targets for 3D point clouds.

Since the pretrained Transformers are tuned as 3D autoencoders, no image, language data, or 3D downstream annotations are required during this cross-modal Transformer transfer. Besides, as the tuned Transformers are only used as the teacher for 3D Transformer student learning, our method does not introduce additional computing or storage costs during downstream feature transferring. Extensive experiments on various tasks have been conducted, which show the superior generalization performance of our ACT pretrained 3D Transformers. For example, an average accuracy improvement of +11.9% is achieved on ScanObjectNN dataset.

To the best of our knowledge, this paper firstly shows that a pretrained foundational Transformer can help 3D representation learning without accessing any 2D, language data, or 3D downstream annotations. ACT is a self-supervised framework that can be generalized to other modalities and tasks, we expect this could spur more exploration of such ACT-style representation learning.

## 2 RELATED WORKS

**Self-Supervised Representation Learning for 3D Geometric Processing** is currently arousing significant interest in the community. Classical methods are built upon reconstruction-based geometry understanding pre-tasks, *e.g.*, point cloud part reordering (Sauder & Sievers, 2019), orientation estimation (Poursaeed et al., 2020), local and global reconstruction (Rao et al., 2020), flow consistency (Mittal et al., 2020), deformation (Achituve et al., 2021), and occlusion (Wang et al., 2021). Concurrently, Xie et al. (2020) propose PointContrast to learn discriminative view consistency between augmented point clouds. Following this direction, various works have been proposed (Zhang et al., 2021; Hou et al., 2021; Chen et al., 2022). Recently, many works have proposed to apply DAE pretraining of point cloud Transformers, and remarkable success has been achieved. Yu et al. (2022) pioneers this direction by extending the idea of BERT-style pretraining (Devlin et al., 2019; Bao et al., 2022), combined with a global contrastive objective (He et al., 2020). Liu et al. (2022a) propose to add some noisy points and classify whether the masked tokens are real or fake for each masked position, which shares a similar pattern with Selfie (Trinh et al., 2019) that classifies whether masked image patches are real or fake. Pang et al. (2022) proposes exploring MAE on point clouds by masked modeling of 3D point cloud coordinates. We follow this DAE-style representation learning paradigm, but different from previous methods, our work seeks to use latent features encoded by the 3D autoencoder with pretrained foundational Transformers as masked modeling targets.

---

[1]For example, the in-house JFT-300M dataset from Google covers over one billion labels for 300M images, and the Common Crawl dataset (Raffel et al., 2020) for NLP consists of nearly one trillion words.

**Cross-Modal 3D Representation Learning** aims at leveraging more modality-inherent learning signals besides 3D point clouds, *e.g.*, 2D images are known to have rich contextual and textural knowledge, while free-form languages are of dense semantics. Mainstream methods are developed upon contrastive learning of global feature matching. For instance, Jing et al. (2021) propose a discriminative Center loss for feature alignment of point clouds, mesh, and images. Afham et al. (2022) propose an intra- and inter-modal contrastive learning framework among augmented point clouds and the corresponding rendered 2D images. By utilizing the geometry prior information for a dense association, another line of work is proposed to explore fine-grained local feature matching. Liu et al. (2021a) propose a contrastive knowledge distillation method to align fine-grained 2D and 3D features. Li et al. (2022) propose a simple contrastive learning framework for inter- and intra- modal dense feature contrast, with the Hungarian algorithm used for better correspondence. Recently, great progress has been made by directly using pretrained 2D image encoders via supervised fine-tuning. Image2Point (Xu et al., 2022) proposes to transfer pretrained weights by convolutional layer inflating. P2P (Wang et al., 2022b) proposes to project 3D point clouds to 2D images as input to the image backbone through a learnable coloring module. Our work also explores whether pretrained foundational models could help 3D learning. However, our method (1) does not use the pretrained 2D or language models as the backbone model for inference, (2) explores using pretrained foundational models from other modalities during self-supervised pretraining without downstream 3D annotations, and (3) does not need the paired point-image or point-language data. Besides 2D images, some works are proposed to utilize natural languages for contrastive 3D representation leanring (Rozenberszki et al., 2022), zero-shot learning (Zhang et al., 2022c), and scene understanding (Zhang et al., 2023).

## 3 PRELIMINARIES

### 3.1 3D POINT CLOUD REPRESENTATIONS WITH TRANSFORMERS

Different from images that lie on regular grids, point clouds are known to be irregular and less structured. Many efforts have been devoted to deep learning architecture design for point cloud data (Qi et al., 2017a;b; Wang et al., 2019), which exploits permutation and translation invariance of a point set for feature learning. Instead of purely relying on such specialized backbones, we leverage the Transformer backbone (Vaswani et al., 2017), which is easier to be unified with other modalities such as image and language and to facilitate cross-modal knowledge transfer. We feed Transformers with local geometry patch embeddings computed using specialized point networks like Qi et al. (2017a) to output more effective geometric representations.

**Local Geometry Patch Embedding** Suppose we have a point cloud $\mathcal{P} = \{\mathbf{p}_i | i = 1, 2, \ldots, N\} \in \mathbb{R}^{N \times 3}$ with $N$ coordinates encoded in a $(x, y, z)$ Cartesian space, we follow Yu et al. (2022) to first sample $N_s$ seed points using farthest point sampling (FPS). The point cloud $\mathcal{P}$ is then grouped into $N_s$ neighborhoods $\mathcal{N} = \{\mathcal{N}_i | i = 1, 2, \ldots, N_s\} \in \mathbb{R}^{N_s \times K \times 3}$ with group centroids from the seed point set $\mathcal{P}_s$. Each neighborhood contains K points generated by searching the K-nearest neighbor of the corresponding seed point. The local geometry feature $\mathbf{x}_i$ around each seed point $\mathbf{p}_i \in \mathcal{P}_s$ is computed by max-pooling per-point features within the neighborhood:

$$\mathbf{x}_i = \underset{\mathbf{p}_{i,j} \in \mathcal{N}_i}{\text{MAX}} \left( \Phi_\theta \left( \xi_{i,j} \right) \right), \tag{1}$$

where $\Phi_\theta(\cdot)$ is a point feature extractor with parameters $\theta$, *e.g.*, per-point MLP as in (Qi et al., 2017a;b), $\xi_{i,j}$ is the feature of $j$-th neighbour point $\mathbf{p}_{i,j}$ in the neighborhood $\mathcal{N}_i$. We will use the set of neighborhood features as token features to feed the following Transformer blocks.

**Transformer Point Feature Encoding** Standard Transformer block (Vaswani et al., 2017) is used as the encoder to further transform local patch embeddings $\mathbf{X} = \{\mathbf{x}_i | i = 1, 2, \ldots, N_s\} \in \mathbb{R}^{N_s \times C}$ with $C$ being the embedding size. Following Yu et al. (2022), we use a two-layer MLP $\psi_\rho$ with learnable parameters $\rho$ as the positional embedding, which is applied to every block for stable training.

$$\mathbf{E}_{\text{pos}} = \left[ \mathbf{E}_{\text{pos}}^{[\text{CLS}]}; \psi_\rho(\mathcal{P}_s) \right], \qquad \mathbf{E}_{\text{pos}}^{[\text{CLS}]} \in \mathbb{R}^C \tag{2}$$

$$\mathbf{h}_0 = [\mathbf{E}^{[\text{CLS}]}; \mathbf{x}_1; \mathbf{x}_2; \cdots; \mathbf{x}_{N_s}] + \mathbf{E}_{\text{pos}}, \qquad \mathbf{E}^{[\text{CLS}]} \in \mathbb{R}^C, \mathbf{E}_{\text{pos}} \in \mathbb{R}^{(N_s+1) \times C} \tag{3}$$

$$\mathbf{h}'_\ell = \text{MSA}\left(\text{LN}(\mathbf{h}_{\ell-1} + \mathbf{E}_{\text{pos}})\right) + \mathbf{h}_{\ell-1}, \qquad \ell = 1 \ldots L \tag{4}$$

$$\mathbf{h}_\ell = \text{MLP}\left(\text{LN}(\mathbf{h}'_\ell)\right) + \mathbf{h}'_\ell, \qquad \ell = 1 \ldots L \tag{5}$$

where MSA denotes alternating layers of multi-head self-attention, LN denotes Layernorm, and MLP is two layers with GELU as non-linearity. $\mathbf{E}^{[\text{CLS}]}$ is a learnable global representation embedding with $\mathbf{E}_{\text{pos}}^{[\text{CLS}]}$ as its learnable positional embedding (Dosovitskiy et al., 2021).

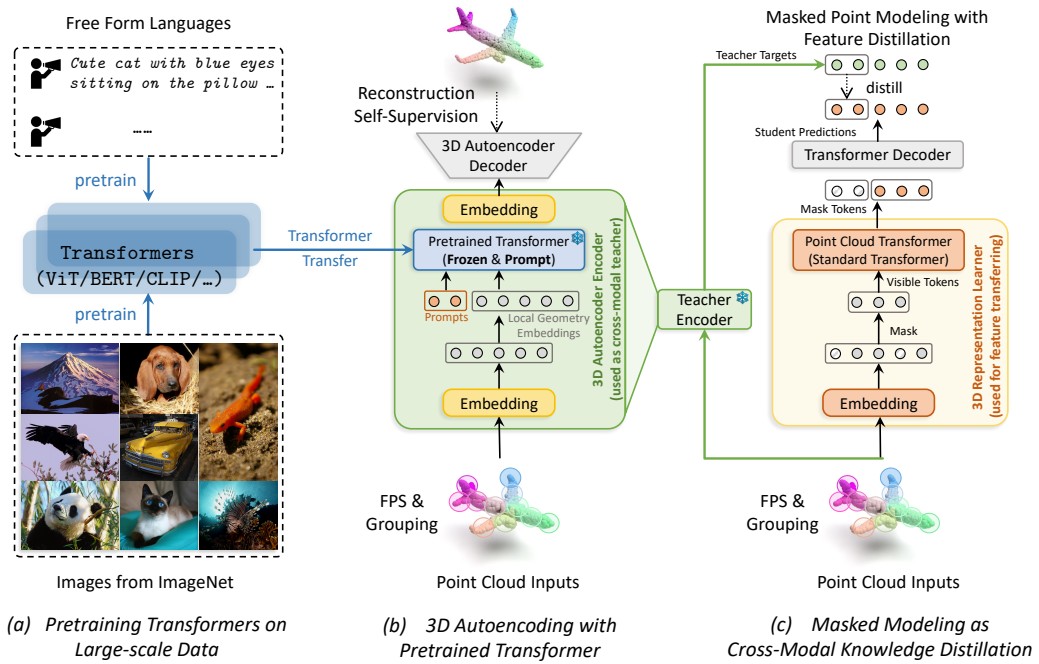

Figure 1: Overview of of our ACT framework (Sec. 3-4). (a) ACT utilizes the Transformers pretrained on large-scale data, *e.g.*, ViT (Dosovitskiy et al., 2021) pretrained with 2D images or BERT (Devlin et al., 2019) pretrained with languages. (b) Stage I of ACT (Sec. 4.1), the pretrained Transformers are tuned by self-supervised 3D autoencoding with prompts (Jia et al., 2022). (c) Stage II of ACT (Sec. 4.2), the 3D autoencoder encoder is used as a cross-modal teacher that encodes latent features as masked point modeling targets for 3D Transformer student representation learning.

## 3.2 KNOWLEDGE DISTILLATION: A UNIFIED VIEW OF MASKED MODELING

Masked signal modeling can be viewed as an extension of the classical denoising autoencoders (DAE) with masked corruption (He et al., 2022b), which has been recently explored for language models (Devlin et al., 2019) and vision (Bao et al., 2022). Formally, given a sequence of $N_t$ tokens $\mathbf{T} = \{\mathbf{t}_i | i = 1, 2, \ldots, N_t\}$, *e.g.*, the token embeddings of an RGB image or point cloud data. The objective is to train a *student* encoder $f_\mathcal{S}$ to predict/reconstruct the output from a *teacher* encoder $f_\mathcal{T}$, where the *teacher* could be a discrete variational autoencoder (dVAE) (Bao et al., 2022) or simply identity mapping (He et al., 2022b). In this fashion, the *student* learns the dark knowledge within data under the guidance of the *teacher*. In order to corrupt the input data, a set of masks $\mathcal{M} = \{m_i | i = 1, 2, \ldots, N_t\} \in \{0, 1\}^{N_t}$ are generated for each position, indicating whether the token is masked or not. A learnable corruption embedding $\boldsymbol{e}^{[\text{M}]}$ is used to replace the masked position, with which the corrupted representation $\mathbf{Z}^\mathcal{M} = \mathbb{1}(\mathcal{M}) \odot \boldsymbol{e}^{[\text{M}]} + \mathbb{1}(1 - \mathcal{M}) \odot \mathbf{T}$ is input to encoder (Devlin et al., 2019) or decoder (He et al., 2022b)[2]. Here, $\odot$ denotes the Hadamard product, and $\mathbb{1}$ is the indicator function. With a distance function $\mathcal{L}_\mathbb{D}(\cdot, \cdot)$ defined in some metric space $\mathbb{D}$ and $h_\mathcal{S}, h_\mathcal{T}$ as the decoders, the objective is to minimize:

$$-\sum_{i=1}^{N_t} m_i \cdot \mathcal{L}_\mathbb{D}\big(h_\mathcal{S} \circ f_\mathcal{S}(\mathbf{Z}^\mathcal{M}), h_\mathcal{T} \circ f_\mathcal{T}(\mathbf{T})\big). \tag{6}$$

The decoders $h$ vary with the modeling targets, *e.g.*, it is a non-linear projection with softmax for BERT (Devlin et al., 2019; Bao et al., 2022) where the metric function becomes Cross-Entropy. Eqn. (6) can be viewed as a unified formulation for masked modeling. It is thus natural to consider how to build a knowledgeable teacher in masked 3D modeling. And our idea is to leverage cross-modal teachers from 2D or language foundation models.

---

[2]For MAE, the encoder only receives visible tokens, and the $\mathbf{T}$ for calculating $\mathbf{Z}^\mathcal{M}$ should be $f_\mathcal{S}\big([t_i | \forall m_i = 0, m_i \in \mathcal{M}]\big)$, where the corrupted representation $\mathbf{Z}^\mathcal{M}$ is fed into the decoder for masked modeling distillation.

# 4 ACT: AUTOENCODERS AS CROSS-MODAL TEACHERS

Our goal is to facilitate 3D representation learning through a pretrained 2D image or language Transformer, which carries dark knowledge absorbed from massive data. However, 3D point clouds are known to have different structures (Li et al., 2022; Afham et al., 2022) from 2D images or languages, which makes the association of fine-grained knowledge difficult. We address this issue by using a two-stage training procedure. An overview of our ACT framework is illustrated in Figure 1.

- **Stage I.** We tune the pretrained 2D or language Transformers as 3D autoencoders, where it learns to understand 3D geometry through self-supervised prompt tuning (Sec. 4.1).
- **Stage II.** We use the pretrained 3D autoencoder as a cross-modal teacher, which is used to distill the latent features to the 3D point cloud Transformer student through masked modeling (Sec. 4.2).

## 4.1 3D AUTOENCODING WITH PRETRAINED FOUNDATIONAL TRANSFORMER

Transformers, recently the dominant architecture in various areas, can model sequential data of any modality in a unified fashion (Vaswani et al., 2017). Therefore, we could directly use the pretrained Transformer blocks by feeding the sequential tokens with 3D positional embeddings of the input point clouds, as described in Sec. 3.1. A lightweight DGCNN is used following Yu et al. (2022), where $\Phi_\theta$ in Eqn. (1) represents the edge convolution layer (Wang et al., 2019).

**Cross-Modal Embedding with Prompts** The point cloud $\mathcal{P}$ is first encoded by the DGCNN-style patch embedding network $g^{\text{pre}}$, producing a set of token embeddings: $\mathbf{X} = g^{\text{pre}}(\mathcal{P})$. Then we prompt the token embeddings and feed them into $D$ layers of pretrained and *frozen* Transformer blocks, *e.g.*, a 2D Transformer $g^{\text{2D}} = \{g_\ell^{\text{2D}} | \ell = 1, 2, ..., D\}$. Here we use $g_\ell^{\text{2D}}$ to denote the $\ell$-th layer of the 2D Transformer. We use $m$ *learnable* prompt embeddings $\mathbf{E}_\ell^{[\text{P}]} = \{e_k^{[\text{P}]} \in \mathbb{R}^C | k \in \mathbb{N}, 1 \leq k \leq m\}$, which are applied to each layer of the Transformer (Jia et al., 2022). Specifically, the $\ell$-th layer $g_l^{\text{2D}}$ of the Transformer transforms the hidden representations $\mathbf{h}_{\ell-1}$ from the $(\ell-1)$-th layer to $\mathbf{h}_\ell$ as below:

$$[\mathbf{h}_\ell; \mathbf{E'}_\ell^{[\text{P}]}] = g_\ell^{\text{2D}}([\mathbf{h}_{\ell-1}; \mathbf{E}_\ell^{[\text{P}]}]), \qquad \ell = 1 \dots D \qquad (7)$$

With this parameter-efficient prompt tuning strategy, we are able to tune the pretrained foundational Transformer while preserving as much pretrained knowledge as possible (He et al., 2022a).

**Point Cloud Autoencoding** Another DGCNN network $g^{\text{post}}$ is used to extract local geometric features from foundational Transformer-embedded hidden representations $\mathbf{h}_\ell$. After this, we leverage a FoldingNet (Yang et al., 2018) to reconstruct the input point cloud. We train the above 3D autoencoder as a discrete variational autoencoder (dVAE) (Kingma & Welling, 2014; Ramesh et al., 2021; Bao et al., 2022) for log-likelihood $\mathrm{P}(p_i|\tilde{p}_i)$ maximization, where $(p_i, \tilde{p}_i) \in \mathcal{D}$ denotes the original and reconstructed point clouds respectively. The overall optimization is to maximize the evidence lower bound (ELBO), which holds when $\beta = 1$ (Ramesh et al., 2021):

$$\sum_{(p_i, \tilde{p}_i) \in \mathcal{D}} \ln \mathrm{P}_\theta(p_i|\tilde{p}_i) \geq \sum_{(p_i, \tilde{p}_i) \in \mathcal{D}} \left( \mathbb{E}_{z_i \sim \mathrm{Q}_\phi(\mathbf{z}|p_i)} \left[ \ln \mathrm{P}_\psi(p_i|z_i) \right] - \beta \mathcal{L}_{\mathbb{KL}} \left[ \mathrm{Q}_\phi(\mathbf{z}|p_i), \mathrm{P}_\psi(\mathbf{z}|\tilde{p}_i) \right] \right),$$

$$(8)$$

where (1) $\mathrm{Q}_\phi(z|p)$ denotes the discrete 3D dVAE tokenizer; (2) $\mathrm{P}_\psi(p|z)$ is the dVAE decoder given discrete point tokens; (3) $\mathrm{P}_\theta(z|\tilde{p})$ reconstructs the input point clouds in an autoencoding way.

## 4.2 MASKED POINT MODELING AS CROSS-MODAL KNOWLEDGE DISTILLATION

By simply training the 3D autoencoder, the strong representation of the pretrained Transformer is translated into the 3D feature space, making the autoencoder spontaneously a cross-modal teacher. We motivate our method with a similar formulation to Eqn. (6). We use the pretrained point cloud encoder introduced in Sec. 4.1 as the teacher $\mathcal{F}_\mathcal{T} = h_\mathcal{T} \circ g^{\text{post}} \circ g^{\text{2D}} \circ g^{\text{pre}}$ and we use a 3D Transformer $\mathcal{F}_\mathcal{S} = h_\mathcal{S} \circ f_\mathcal{S}$ as the student. The masked point modeling as cross-modal knowledge distillation minimizes a negative cosine similarity $\mathcal{L}_{\cos}(\mathbf{s}, \mathbf{t}) = 1 - \frac{\mathbf{s} \cdot \mathbf{t}}{\|\mathbf{s}\| \cdot \|\mathbf{t}\|}$ between the encoded teacher and student features:

$$-\sum_{i=1}^{N_t} m_i \cdot \mathcal{L}_{\cos}(\mathcal{F}_\mathcal{S}(\mathbf{Z}^\mathcal{M}), \mathcal{F}_\mathcal{T}(\mathbf{T})). \qquad (9)$$

Table 2: Classification results on ScanObjectNN. Ours[1]: results trained with no data augmentation. Ours[2]: results trained with simple point cloud rotation. DA: data augmentation is used during fine-tuning training. The overall accuracy, *i.e.*, OA (%) is reported.

| Method | #Params(M) | DA | OBJ_BG | OBJ_ONLY | PB_T50_RS |
|---|---|---|---|---|---|
| *Supervised Learning Only* | | | | | |
| PointNet (Qi et al., 2017a) | 3.5 | ✓ | 73.3 | 79.2 | 68.0 |
| SpiderCNN (Xu et al., 2018) | - | ✓ | 77.1 | 79.5 | 73.7 |
| PointNet++ (Qi et al., 2017b) | 1.5 | ✓ | 82.3 | 84.3 | 77.9 |
| DGCNN (Wang et al., 2019) | 1.8 | ✓ | 82.8 | 86.2 | 78.1 |
| PointCNN (Li et al., 2018) | 0.6 | ✓ | 86.1 | 85.5 | 78.5 |
| BGA-DGCNN (Uy et al., 2019a) | 1.8 | ✓ | - | - | 79.7 |
| BGA-PN++ (Uy et al., 2019a) | 1.5 | ✓ | - | - | 80.2 |
| DRNet (Qiu et al., 2021) | - | ✓ | - | - | 80.3 |
| GBNet (Qiu et al., 2022) | 8.8 | ✓ | - | - | 80.5 |
| SimpleView (Goyal et al., 2021) | - | ✓ | - | - | 80.5±0.3 |
| PRANet (Cheng et al., 2021) | 2.3 | ✓ | - | - | 81.0 |
| MVTN (Hamdi et al., 2021) | - | ✓ | - | - | 82.8 |
| PointMLP (Ma et al., 2022) | 13.2 | ✓ | - | - | 85.4±0.3 |
| *with Self-Supervised Representation Learning* (FULL) | | | | | |
| Transformer (Vaswani et al., 2017) | 22.1 | ✓ | 79.86 | 80.55 | 77.24 |
| OcCo (Wang et al., 2021) | 22.1 | ✓ | 84.85 | 85.54 | 78.79 |
| Point-BERT (Yu et al., 2022) | 22.1 | ✓ | 87.43 | 88.12 | 83.07 |
| MaskPoint (Liu et al., 2022a) | 22.1 | ✓ | 89.30 | 88.10 | 84.30 |
| Point-MAE (Pang et al., 2022) | 22.1 | ✓ | 90.02 | 88.29 | 85.18 |
| ACT (Ours[1]) | 22.1 | × | **91.22** | **89.16** | **85.81** |
| ACT (Ours[2]) | 22.1 | ✓ | **93.29** | **91.91** | **88.21** |
| Point-MAE (Pang et al., 2022) | 22.1 | ✓ | 89.31±0.41 | 87.88±0.36 | 84.35±0.31 |
| ACT (Ours[1]) | 22.1 | × | **90.06**±0.56 | **89.02**±0.22 | **85.33**±0.27 |
| ACT (Ours[2]) | 22.1 | ✓ | **92.48**±0.59 | **91.57**±0.37 | **87.88**±0.36 |
| *with Self-Supervised Representation Learning* (MLP-LINEAR) | | | | | |
| Point-MAE (Pang et al., 2022) | 22.1 | ✓ | 82.58±0.58 | 83.52±0.41 | 73.08±0.30 |
| ACT (Ours[1]) | 22.1 | × | **82.71**±0.45 | **84.34**±0.29 | **74.17**±0.05 |
| ACT (Ours[2]) | 22.1 | ✓ | **85.20**±0.83 | **85.84**±0.15 | **76.31**±0.26 |
| *with Self-Supervised Representation Learning* (MLP-3) | | | | | |
| Point-MAE (Pang et al., 2022) | 22.1 | ✓ | 84.29±0.55 | 85.24±0.67 | 77.34±0.12 |
| ACT (Ours[1]) | 22.1 | × | **85.67**±0.29 | **86.79**±0.30 | **78.89**±0.22 |
| ACT (Ours[2]) | 22.1 | ✓ | **87.14**±0.22 | **88.90**±0.40 | **81.52**±0.19 |

## 5 EXPERIMENTS

### 5.1 TRANSFER LEARNING ON DOWNSTREAM TASKS

**Transfer Protocol** We use three variants of transfer learning protocols for classification tasks:

(a) FULL: Fine-tuning pretrained models by updating *all* backbone and classification heads.

(b) MLP-LINEAR: The classification head is a single-layer linear MLP, and we only update this head parameters during fine-tuning.

(c) MLP-3: The classification head is a three-layer non-linear MLP (which is the same as the one used in FULL), and we only update this head parameters during fine-tuning.

**3D Real-world Dataset Classification** We first show the evaluation of 3D shape recognition on the challenging real-world dataset ScanObjectNN (Uy et al., 2019b). The results are shown in Table 2, where it is observed that: (i) Comparing to Transformer *from scratch* baseline under FULL tuning protocol, our ACT gains a significant improvement of +10.4% accuracy averaged on the three variant ScanObjectNN benchmarks. Further, with simple point cloud rotation, ACT achieves an average improvement of +11.9%; (ii) In comparison to methods explicitly designed with 3D geometry understanding purpose, our ACT achieves consistently better results. (iii) Compared to other self-supervised learning (SSL) methods, our ACT achieves the best generalization across all

Table 3: Classification results on the ModelNet40 dataset. The overall accuracy, *i.e.*, OA (%) is reported. [ST]: standard Transformer architecture.

| Method | [ST] | #Point | OA (%) |
|---|---|---|---|
| *Supervised Learning Only* | | | |
| PointNet (Qi et al., 2017a) | - | 1k P | 89.2 |
| PointNet++ (Qi et al., 2017b) | - | 1k P | 90.7 |
| PointNet++ (Qi et al., 2017b) | - | 5k P+N | 91.9 |
| PointCNN (Li et al., 2018) | - | 1k P | 92.5 |
| PointConv (Wu et al., 2019) | - | 1k P+N | 92.5 |
| KPConv (Thomas et al., 2019) | - | 1k P | 92.9 |
| DGCNN (Wang et al., 2019) | - | 1k P | 92.9 |
| RS-CNN (Liu et al., 2019b) | - | 1k P | 92.9 |
| DensePoint (Liu et al., 2019a) | - | 1k P | 93.2 |
| PointASNL (Yan et al., 2020) | - | 1k P | 92.9 |
| PosPool (Liu et al., 2020) | - | 5k P | 93.2 |
| DRNet (Qiu et al., 2021) | - | 1k P | 93.1 |
| Point Trans. (Engel et al., 2020) | × | 1k P | 92.8 |
| PCT (Guo et al., 2021) | × | 1k P | 93.2 |
| PointTransformer (Zhao et al., 2021) | × | 1k P | 93.7 |
| NPCT (Guo et al., 2021) | ✓ | 1k P | 91.0 |

| Method | [ST] | #Point | OA (%) |
|---|---|---|---|
| *with Self-Supervised Representation Learning* (FULL) | | | |
| Transformer (Vaswani et al., 2017) | ✓ | 1k P | 91.4 |
| Transformer (Vaswani et al., 2017) | ✓ | 4k P | 91.2 |
| OcCo (Wang et al., 2021) | ✓ | 1k P | 92.1 |
| OcCo (Wang et al., 2021) | ✓ | 4k P | 92.2 |
| Point-BERT (Yu et al., 2022) | ✓ | 1k P | 93.2 |
| Point-MAE (Pang et al., 2022) | ✓ | 1k P | **93.8** |
| ACT (Ours) | ✓ | 1k P | **93.7** |
| Point-MAE (Pang et al., 2022) | ✓ | 1k P | 93.12±0.25 |
| ACT (Ours) | ✓ | 1k P | **93.50±0.08** |
| *with Self-Supervised Representation Learning* (MLP-LINEAR) | | | |
| Point-MAE (Pang et al., 2022) | ✓ | 1k P | 91.22±0.26 |
| ACT (Ours) | ✓ | 1k P | **91.36±0.17** |
| *with Self-Supervised Representation Learning* (MLP-3) | | | |
| Point-MAE (Pang et al., 2022) | ✓ | 1k P | 92.33±0.09 |
| ACT (Ours) | ✓ | 1k P | **92.69±0.18** |

transferring protocols on ScanObjectNN. Besides, ACT succeeds in reaching the state-of-the-art (SOTA) performance among methods using pure 3D Transformer architecture on ScanObjectNN, *e.g.*, ACT outperforms Point-MAE by +3.0% accuracy on the most challenging PB_T50_RS benchmark.

**3D Scene Segmentation** Semantic segmentation on large-scale 3D scenes is challenging, demonstrating the understanding of contextual semantics and local geometric relationships. In Table 4, we report the results on S3DIS dataset (Armeni et al., 2016). It can be seen that: (i) ACT significantly improves the *from scratch* baseline by +2.5% and +1.2% mAcc and mIoU, respectively. (ii) ACT outperforms the SSL counterpart Point-MAE by +1.2% and +0.4% mAcc and mIoU, showing superior transferring capacity on the large-scene dataset. (iii) With only geometric inputs $xyz$, ACT can achieve comparable or better performance to architectures with the meticulous design using $xyz+rgb$ data, including 3D-specific Transformer architecture (Guo et al., 2021).

Table 4: Semantic segmentation results on the S3DIS Area 5. The mean accuracy and mean IoU across all categories, *i.e.*, mAcc (%) and mIoU (%) are reported. $xyz$: point cloud coordinates are used. $xyz+rgb$: both coordinates and RGB color are used.

| Methods | Input | mAcc (%) | mIoU (%) |
|---|---|---|---|
| PointNet | $xyz+rgb$ | 49.0 | 41.1 |
| PointNet++ | $xyz+rgb$ | 67.1 | 53.5 |
| PointCNN | $xyz+rgb$ | 63.9 | 57.3 |
| PCT | $xyz+rgb$ | 67.7 | 61.3 |
| Transformer | $xyz$ | 68.6 | 60.0 |
| Point-MAE | $xyz$ | 69.9 | 60.8 |
| ACT (Ours) | $xyz$ | **71.1** | **61.2** |

**3D Synthetic Dataset Classification** We show the evaluation of 3D shape classification on synthetic dataset ModelNet40 (Wu et al., 2015). To demonstrate the data-efficiency property of ACT given limited training examples, we first follow Sharma & Kaul (2020) to evaluate few-shot learning. From Table 5, we see: (i) ACT brings significant improvements of +9.0%, +4.7%, +8.7%, +6.2% respectively for the four settings over *from scratch* FULL transferring baseline. (ii) Our ACT consistently achieves the best performance compared to other SSL methods. Then, we show results on the full dataset in Table 3, where we observe that our ACT achieves a +2.5% accuracy improvement compared to the *from scratch* baseline under FULL protocol, and the results are comparable or better to other self-supervised learning methods across all transferring protocols.

Table 5: Few-shot classification on ModelNet40, overall accuracy (%) is reported.

| Method | 5-way | | 10-way | |
|---|---|---|---|---|
| | 10-shot | 20-shot | 10-shot | 20-shot |
| DGCNN | 31.6 ± 2.8 | 40.8 ± 4.6 | 19.9 ± 2.1 | 16.9 ± 1.5 |
| OcCo | 90.6 ± 2.8 | 92.5 ± 1.9 | 82.9 ± 1.3 | 86.5 ± 2.2 |
| *with Self-Supervised Representation Learning* (FULL) | | | | |
| Transformer | 87.8 ± 5.2 | 93.3 ± 4.3 | 84.6 ± 5.5 | 89.4 ± 6.3 |
| OcCo | 94.0 ± 3.6 | 95.9 ± 2.3 | 89.4 ± 5.1 | 92.4 ± 4.6 |
| Point-BERT | 94.6 ± 3.1 | 96.3 ± 2.7 | 91.0 ± 5.4 | 92.7 ± 5.1 |
| Point-MAE | 96.3 ± 2.5 | 97.8 ± 1.8 | 92.6 ± 4.1 | 95.0 ± 3.0 |
| ACT (Ours) | **96.8 ± 2.3** | **98.0 ± 1.4** | **93.3 ± 4.0** | **95.6 ± 2.8** |
| *with Self-Supervised Representation Learning* (MLP-LINEAR) | | | | |
| Point-MAE | 91.1 ± 5.6 | 91.7 ± 4.0 | 83.5 ± **6.1** | 89.7 ± **4.1** |
| ACT (Ours) | **91.8 ± 4.7** | **93.1 ± 4.2** | **84.5** ± 6.4 | **90.7** ± 4.3 |
| *with Self-Supervised Representation Learning* (MLP-3) | | | | |
| Point-MAE | 95.0 ± 2.8 | 96.7 ± 2.4 | 90.6 ± **4.7** | 93.8 ± 5.0 |
| ACT (Ours) | **95.9 ± 2.2** | **97.7 ± 1.8** | **92.4** ± 5.0 | **94.7 ± 3.9** |

Table 6: Ablation study on the depth of the pretraining decoder.

| Dec. Depth | OA (%)↑ |
|------------|---------|
| 0 | 83.69 |
| 1 | 85.11 |
| 2 | **85.33** |
| 4 | 84.98 |

Figure 2: Ablation study of masking ratio and cross-modal Transformer teacher choice.

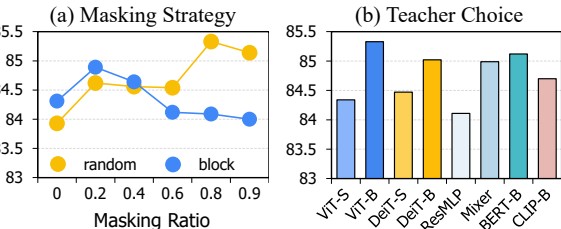

Table 7: Ablation study on different training strategies of the dVAE tokenizer. The F-Score, Chamfer distance using L1-norm and L2-norm, *i.e.*, CD-$\ell_1$ and CD-$\ell_2$ are reported.

| Methods | Num. of Prompt | Prompt Type | Freeze | F-Score↑ | CD-$\ell_1$ ↓ | CD-$\ell_2$ ↓ |
|---------|----------------|-------------|--------|----------|---------------|---------------|
| Point-BERT dVAE | N/A | N/A | N/A | 0.166 | 25.933 | 2.697 |
| DeiT-B dVAE | 0 | N/A | × | 0.175 | 24.589 | 2.380 |
| DeiT-B dVAE | 0 | N/A | ✓ | 0.180 | 24.090 | 2.274 |
| DeiT-B dVAE | 32 | shallow | ✓ | 0.188 | 23.769 | 2.196 |
| DeiT-B dVAE | 32 | deep | ✓ | 0.189 | 23.873 | 2.173 |
| DeiT-B dVAE | 64 | deep | ✓ | 0.189 | 23.229 | 2.127 |
| ViT-B dVAE | 64 | deep | ✓ | 0.193 | 23.524 | 2.110 |

## 5.2 ABLATION STUDY

**Decoder Depth** Table 6 shows the average fine-tuning accuracy on ScanObjectNN using ACT with different depths of decoders. It can be seen that the performance is not sensitive to the decoder depth, and we find that decoder with 2 blokcs achieves the highest results. Note that when decoder depth is 0, we adopt a masked modeling architecture similar to BERT (Devlin et al., 2019), where there is no decoder, and the encoder sees all tokens, including masked ones. We find that this leads to an inferior result, consistent with the observation in 2D that data of low semantics requires a non-trivial decoder for modeling purpose (He et al., 2022b).

**Masking Strategy and Teacher Choice** Figure 2(a) shows the average fine-tuning on ScanobjectNN with different masking strategies. It can be observed that a higher masking ratio using random masking yields better results, while block masking has an appetite for lower masking ratios. Note that when the masking ratio is zero, we use vanilla knowledge distillation for all tokens, and it leads to inferior performance. Figure 2(b) shows average fine-tuning accuracy on ScanObjectNN using ACT with different teacher Transformers including Vision Transformers (Dosovitskiy et al., 2021; Touvron et al., 2021b), all-MLP architectures (Tolstikhin et al., 2021; Touvron et al., 2021a), language model (Devlin et al., 2019) and vision-language model (Radford et al., 2021). It is observed that a larger teacher consistently yields better performance. Moreover, surprisingly, our ACT with language model BERT-B (*i.e.*, BERT$_{base}$) as the cross-modal teacher can achieve an average accuracy of 85.12±0.54% (up to 85.88%), demonstrating that ACT can generalize to any modality.

**3D Autoencoder Training** Table 7 shows the reconstruction results of different training configurations for the 3D autoencoder with a pretrained 2D image Transformer. It is observed that: (i) Our 3D dVAE model with pretrained image Transformer achieves significantly better reconstruction results than Point-BERT. It demonstrates that the pretrained 2D image Transformers have a strong representation capacity for 3D. (ii) Prompt tuning or freezing the model leads to better results than full tuning, and we argue that it is because some pretrained 2D knowledge is forgotten, and prompt tuning effectively addresses this issue. Reconstruction visualizations can be found in Appendix D.

## 6 DISCUSSIONS

### 6.1 IS A STRONGER TOKENIZER ALL YOU NEED?

In order to understand the necessity of the pretrained 2D image Transformer in the 3D dVAE model, we have conducted experiments with different dVAE teachers and masked modeling configurations. From Table 8, we see that: (i) When using the Point-BERT dVAE model without pretrained 2D image Transformers, by distilling the latent feature instead of discrete tokens, we can achieve +0.62% improvement. Our analysis agrees that discrete token identification is more challenging to learn for

Table 8: Study on the effect of pretrained image Transformer-based 3D Autoencoder.

| Teacher | Target | OA (%)↑ |
|---|---|---|
| Point-BERT | Point-BERT | 83.07 |
| Point-BERT | Ours | 83.69 |
| Ours | Point-BERT | 82.51 |
| Ours | Ours | **85.81** |

Table 9: Study of applying our method as auxiliary knowledge distillation during pretraining.

| Method | KD | OA (%)↑ |
|---|---|---|
| Point-MAE | × | 85.18 |
| Our Impl. | ✓ | **86.05** |
| Our Impl. | × | 84.35±0.31 |
| Our Impl. | ✓ | **84.96±0.58** |

Table 10: Study of different positional embeddings for 2D image transformer in dVAE model. (a) N/A: no positional embedding is used. (b) 2D/$z$: positional embedding with only 2D $xy$ plane coordinates. (c) 3D: positional embedding with all 3D $xyz$ coordinates. The F-Score, Chamfer distance using L1-norm and L2-norm, *i.e.*, CD-$\ell_1$ and CD-$\ell_2$, and OA on ScanObjectNN are reported.

| Methods | pos embed | F-Score↑ | CD-$\ell_1$ ↓ | CD-$\ell_2$ ↓ | OA (%)↑ |
|---|---|---|---|---|---|
| ViT-B dVAE | N/A | 0.166 | 25.918 | 2.698 | 84.21±0.45 |
| ViT-B dVAE | 2D/$z$ | 0.184 | 24.135 | 2.259 | 85.10±0.45 |
| ViT-B dVAE | 3D | 0.193 | 23.524 | 2.110 | 85.33±0.27 |

3D data. (ii) When using Point-BERT discrete token as the masked modeling target, by applying our dVAE model with pretrained 2D image Transformers, we get the worst performance. It demonstrates that the discrete tokens are not suitable for the semantically sparse point cloud data, no matter how strong the tokenizer is. (iii) When using our ACT, the performance is significantly improved. It demonstrates that the 3D dVAE with pretrained 2D image Transformer can encode features with rich semantics, which is better suited for masked point modeling.

## 6.2 Can ACT be used as an auxiliary knowledge distillation method?

Since our ACT uses encoded features as masked modeling targets, it brings another potential to apply our method as auxiliary feature distillation. Table 9 shows the results of training Point-MAE with ACT as auxiliary deep supervision of the intermediate features, where the ACT encoded latent features are distilled to the encoder feature of Point-MAE. We can observe that ACT can improve Point-MAE significantly by +0.87% of accuracy on ScanObjectNN, demonstrating that ACT is scalable and effective as a knowledge distillation method.

## 6.3 How does the 2D Vision Transformer understand 3D point clouds?

To better understand how the 2D image Transformers understand 3D inputs through the autoencoder training, we study the effect of positional embedding used by ViT-B in our ACT dVAE model. From Table 10, we can observe that: (i) Without any positional embedding, the pretrained ViT still learns transferable 3D features (84.21±0.45% accuracy). We argue that it is because the positional geometry information is already contained in the input 3D coordinates and the pretrained 2D Transformer can process 3D data purely by geometry features without explicit positional hints. (ii) When using positional embedding with only 2D $xy$ plane coordinates, accuracy is improved significantly by +0.89%. We argue that 2D positional embedding is learned to fit the frozen image Transformer, enabling the image Transformer to encode 3D inputs into pretrained 2D feature space with high semantics. (iii) With all 3D coordinates used for positional embedding, the 2D image Transformer succeeds in leveraging the additional coordinate information for better feature encoding.

## 7 Conclusions

This paper presents a self-supervised learning framework ACT that performs masked modeling as feature distillation from pretrained foundational Transformers to 3D Transformer students. ACT first transfers the pretrained foundational Transformers as cross-modal 3D teachers via self-supervised 3D autoencoding. The semantic-enriched latent feature from the tuned 3D autoencoder is then used as masked modeling targets for the 3D Transformer students' representation learning, which shows remarkable generalization performance over various downstream 3D tasks. As a general SSL framework, we believe ACT could be easily extended to other modalities than 3D data. A great potential is shown to transfer cross-modal knowledge in this self-supervised fashion, which may largely facilitate the development of foundational modeling in this data-driven deep learning era.

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

## A    ADDITIONAL RELATED WORKS

**Self-Supervised Representation Learning**   has achieved remarkable success in both natural language processing (Devlin et al., 2019; Brown et al., 2020) and 2D visual understanding (Noroozi & Favaro, 2016; Dosovitskiy et al., 2016; Pathak et al., 2016; Ye et al., 2019). One prominent strand of research follows the contrastive objective via *construct, then contrast* for learning constructed invariance and consistency (Hadsell et al., 2006; Wu et al., 2018; van den Oord et al., 2018; Hjelm et al., 2019; Chuang et al., 2020; Grill et al., 2020; Chen et al., 2020b; He et al., 2020; Chen & He, 2021; Zhang et al., 2022a). Another paradigm lies in training denoising autoencoders (DAE) (Vincent et al., 2008; 2010) via *corrupt, then reconstruct (predict)* data signals in a self-supervised fashion. With rapid development of Transformers in vision (Vaswani et al., 2017; Dosovitskiy et al., 2021; Liu et al., 2021b), abundant works have been proposed to generalize DAE to masked modeling of RGB pixel (Zhang et al., 2016; Chen et al., 2020a; He et al., 2022b), pretrained DALL-E token (Ramesh et al., 2021; Bao et al., 2022), online teacher token feature (Zhou et al., 2022), and HOG feature (Dalal & Triggs, 2005; Wei et al., 2022). Recently, the exploration of combining the merits of these two paradigms has been proposed by several works (Tian et al., 2022; Yi et al., 2022; Tao et al., 2022).

**Knowledge Distillation**   generally requires training of the student model to mimic the knowledgeable teacher, in which the dark knowledge is transferred. This technique was first proposed by Bucila et al. (2006) for model compression purposes, which is further extended by Hinton et al. (2015) for deep neural networks. Afterwards, it becomes a most utilized technique for model compression in 2D vision (Romero et al., 2015; Zagoruyko & Komodakis, 2017; Zhang & Ma, 2021), natural language processing (Sanh et al., 2019; Jiao et al., 2020) and 3D vision (Zhang et al., 2022b; Yang et al., 2022). Recently, this technique has been extended for efficient visual representation learning through self-distillation (Zhang et al., 2019) of distillation token (Touvron et al., 2021b) or momentum tokenizer feature (Zhou et al., 2022).

## B    IMPLEMENTATION DETAILS

### B.1    SELF-SUPERVISED PRETRAINING SETUP

**Data**   We use ShapeNetCore from ShapeNet (Chang et al., 2015) as the pretraining dataset. ShapeNet is a collection of clean 3D CAD object models with rich annotations consisting of ∼51K unique 3D models from 55 common object categories. We sample 1,024 points per 3D model sample using farthest point sampling (FPS), which is further divided into 64 groups of 32 points as local geometry patches using KNN. Standard data augmentations are adopted during pretraining the 3D autoencoder and 3D point cloud Transformer, *i.e.*, random scaling and translation.

**3D Autoencoder**   Following Yu et al. (2022), we use a lightweight DGCNN (Wang et al., 2019) as the local geometry patch embedding module, which takes the KNN groups as input and models the local geometry relationship through dynamic graph message passing. The encoded geometry patch embedding is then fed into a pretrained 2D image Transformer, *e.g.*, ViT (Dosovitskiy et al., 2021) or DeiT (Touvron et al., 2021b). Note that without specific descriptions, the results in the paper use ViT-B pretrained on ImageNet (Deng et al., 2009) as the 2D image Transformer. Besides, only the Transformer blocks and layer normalization are used while other layers like original 2D patch embedding are dropped. The decoder is several DGCNN layers to further model 2D-embedded 3D features, followed by the FoldingNet (Yang et al., 2018) for autoencoder reconstruction. As pointed out by Ramesh et al. (2021), the weight of the KL divergence loss (*i.e.*, $\beta$ in Eqn. (8)) during training must be small, we also set the KL divergence loss to 0 in the first 10K steps which is gradually increased to 0.1 in the following 100K steps. We use AdamW optimizer (Loshchilov & Hutter, 2019) with a learning rate 5e-4. The cosine learning rate scheduler is adopted with 60K warming-up steps. Following Chen et al. (2020a), The Gumbel-softmax temperature decayed from 1 to 0.0625 in 100K steps. The batch size is set to 64, and the overall training includes ∼150K steps.

The training of the 3D autoencoder is supervised by the reconstruction objective and the variational distribution loss. Following Yu et al. (2021), we use coarse- and fine-grained predictions with the

ground-truth point cloud. The $\ell_1$-stle Chamfer Distance is used as the reconstruction objective:

$$\mathcal{L}_{CD-\ell_1}(\mathcal{P},\mathcal{G}) = \frac{1}{|\mathcal{P}|}\sum_{p\in\mathcal{P}}\min_{g\in\mathcal{G}}\|p-g\| + \frac{1}{|\mathcal{G}|}\sum_{g\in\mathcal{G}}\min_{p\in\mathcal{P}}\|g-p\|, \tag{10}$$

where $\mathcal{P}$ denotes the predicted point clouds and $\mathcal{G}$ denotes the ground-truth point clouds. Following Ramesh et al. (2021), we use a uniform prior for the discrete variational autoencoder (dVAE) training, where the KL-divergence is adopted for distribution alignment. Hence, the overall objective function is:

$$\mathcal{L}_{\text{dVAE}} = \mathcal{L}_{CD-\ell_1}(\mathcal{P}_{\text{coarse}},\mathcal{G}) + \mathcal{L}_{CD-\ell_1}(\mathcal{P}_{\text{fine}},\mathcal{G}) + \beta\mathcal{L}_{\mathbb{KL}}. \tag{11}$$

**Masked Point Modeling**   For masked point modeling, the autoencoder encoder as the backbone model is a standard Transformer architecture (Vaswani et al., 2017) with a lightweight PointNet (Qi et al., 2017a) patch embedding module, and the decoder is also a Transformer architecture. The encoder Transformer has 12 blocks with an embedding dimension set to 384, while the decoder Transformer has only 2 blocks with the same embedding dimension. The multi-head attention in the Transformer has 6 heads, and the MLP ratio is set to 4. Stochastic depth (Huang et al., 2016) with rate 0.1 is applied to all Transformer blocks. The AdamW optimizer is adopted with a cosine learning rate of 1e-3 and a weight decay of 5e-2. The model is pretrained for 300 epochs with a batch size of 128.

### B.2 TRANSFER LEARNING SETUP

**ModelNet40**   ModelNet40 (Wu et al., 2015), as one of the most classical datasets, is used for the evaluation of object classification on clean 3D CAD models. There are ∼12K meshed 3D CAD models covering 40 categories. For benchmarking purposes, we use the standard data split of 9,843/2,468 respectively for training and validation, following Qi et al. (2017b). The classification head is a three-layer MLP with a dropout of 0.5, and the hidden layer dimension is set to 384, the same as the Transformer backbone. AdamW optimizer with a 0.05 weight decay is used. Cosine learning rate scheduler is used with a 5e-4 learning rate, warming up 10 epochs. The batch size is 32, and the total training is 300 epochs. Standard random scaling and translation augmentations are used and note that we use a voting-based evaluation strategy (Liu et al., 2019b) for a fair comparison.

**ScanObjectNN**   ScanObjectNN dataset (Uy et al., 2019b) is a collection of 3D object point clouds from the challenging real-world indoor scene ScanNet dataset (Dai et al., 2017), which includes ∼15K objects from 15 categories. We use three variants of ScanObjectNN following Uy et al. (2019b), *i.e.*, OBJ_BG, OBJ_ONLY, and PB_T50_RS. The optimization and other training settings (*e.g.*, training epochs) are the same with ModelNet40. For data augmentations, we report results trained with no data augmentations and simple point cloud rotation as used by Wang et al. (2022b). Note that no voting strategy is adopted during testing, and if without a specific description, we report overall accuracy (OA) on the most challenging PB_T50_RS benchmark.

**ShapeNetPart**   ShapeNetPart dataset (Yi et al., 2016) is a popular point-level synthetic object part segmentation benchmark, which covers ∼17K objects from 16 object categories with 50 fine-grained part categories. We use AdamW optimizer with 1e-5 weight decay. Cosine Learning rate 2e-5 with 10 epochs warming up is used. Standard random scaling and translation are used as a data augmentation strategy. The batch size is set to 16, and we train models for 300 epochs.

**S3DIS**   S3DIS dataset (Armeni et al., 2016) provides densely annotated semantic labels for point clouds. It is consisted of six large-scale indoor areas from three different buildings, covering a total of 273 million points from 13 categories. Following Tchapmi et al. (2017), we advocate using Area 5 for evaluation purposes for better and fair generalization performance benchmarking. We use AdamW optimizer with 1e-5 weight decay, with a cosine learning rate of 2e-5 warming up to 10 epochs. The batch size is 32, and the total training involves 60 epochs.

**ScanNetV2**   ScanNetV2 (Dai et al., 2017) is a large-scale dataset that collects ∼2.5M RGB-D scans from 1,513 indoor scenes with comprehensive annotations. Following Liu et al. (2022a), we construct a *ScanNet-Medium* subset containing ∼15K frames with a sampling rate of 100 from the raw dataset for 300 epochs ACT pretraining. We use 3DETR (Misra et al., 2021) with the same training recipe for 3D object detection downstream transferring. Note that only the encoder is pretrained and transferred, which has 3 layers with an embedding dimension of 384, and the decoder has 8 layers.

## C ADDITIONAL EXPERIMENTS

**3D Object Detection** We evaluate the representation capability of ACT with downstream 3D object detection on large-scale scene dataset ScanNetV2 with 3DETR (Misra et al., 2021). From Table 11, it is observed that (i) ACT significantly improves by +1.7% $AP_{25}$ and +4.2% $AP_{50}$ to the *from scratch* baseline. (ii) In comparison to other SSL methods, ACT outperforms MaskPoint by a clear margin.

Table 11: 3D object detection on the ScanNetV2 dataset. The detection performance using mean Average Precision (mAP) at two different IoU thresholds of 0.50 and 0.25, *i.e.*, $AP_{50}$ and $AP_{25}$ are reported. $xyz$: point cloud coordinates are used.

| Method | SSL | Input | $AP_{50}$ | $AP_{25}$ |
|---|---|---|---|---|
| VoteNet (Qi et al., 2019) | × | $xyz$ | 33.5 | 58.6 |
| PointContrast (Xie et al., 2020) | ✓ | $xyz$ | 38.0 | 59.2 |
| STRL (Huang et al., 2021) | ✓ | $xyz$ | 38.4 | 59.5 |
| RandomRooms (Rao et al., 2021) | ✓ | $xyz$ | 36.2 | 61.3 |
| DepthContrast (Zhang et al., 2021) | ✓ | $xyz$ | - | 61.3 |
| 3DETR (Misra et al., 2021) | × | $xyz$ | 37.9 | 62.1 |
| Point-BERT (Yu et al., 2022) | ✓ | $xyz$ | 38.3 | 61.0 |
| MaskPoint (Liu et al., 2022a) | ✓ | $xyz$ | 40.6 | 63.4 |
| ACT (Ours) | ✓ | $xyz$ | **42.1** | **63.8** |

**Comparison to Supervised Cross-Modal 3D Representation Learning Methods** Table 12 shows the comparison of our method to the cross-modal 3D representation learning method P2P (Wang et al., 2022b) that also uses extra image data by supervised fine-tuning of the pretrained image models. From the results, it is observed that our ACT achieves 88.21% OA on PB_T50_RS with only 22.1M pure 3D Transformer, while P2P achieves 87.4%/89.3% with 42.7M/195.8M large-scale image models (*i.e.*, ResNets101 (He et al., 2016) and HorNet (Rao et al., 2022)).

Table 12: Comparison to supervised cross-modal 3D representation learning method on ScanObjectNN. Overall accuracy, *i.e.*, OA (%) is reported.

| Method | Backbone | #Params (M) | OA (%)↑ |
|---|---|---|---|
| P2P (Wang et al., 2022b) | ResNet101 | 42.7 | 87.4 |
| P2P (Wang et al., 2022b) | HorNet | 195.8 | 89.3 |
| ACT (Ours) | Transformer | 22.1 | 88.2 |

**3D Part Segmentation** ShapeNetPart (Yi et al., 2016) is used to evaluate the learning capacity toward knowledge of detailed shape semantics within 3D objects. Table 13 shows the detailed IoU results of every category, from which we see: (i) ACT significantly improves the *from scratch* baseline by 1.2% and 1.0% of Cls. mIoU and Ins. mIoU, respectively. (ii) ACT outperforms the other methods, achieving up to 12 top or second IoU performances over the total 16 categories.

Table 13: Part segmentation results on the ShapeNetPart dataset. The mean IoU across all categories, *i.e.*, Cls. mIoU, the mean IoU across all instances, *i.e.*, Ins. mIoU (%), and IoU (%) for each category are reported. The best results are **bolded** and the second best results are underlined.

| Method | Cls. mIoU | Ins. mIoU | aero | bag | cap | car | chair | aerp-hone | guitar | knife | lamp | laptop | motor-bike | mug | pistol | rocket | skate-board | table |
|---|---|---|---|---|---|---|---|---|---|---|---|---|---|---|---|---|---|---|
| PointNet | 80.39 | 83.7 | 83.4 | 78.7 | 82.5 | 74.9 | 89.6 | 73.0 | 91.5 | 85.9 | 80.8 | 95.3 | 65.2 | 93.0 | 81.2 | 57.9 | 72.8 | 80.6 |
| PointNet++ | 81.85 | 85.1 | 82.4 | 79.0 | 87.7 | 77.3 | 90.8 | 71.8 | 91.0 | 85.9 | 83.7 | 95.3 | 71.6 | 94.1 | 81.3 | 58.7 | 76.4 | 82.6 |
| DGCNN | 82.33 | 85.2 | 84.0 | 83.4 | 86.7 | 77.8 | 90.6 | 74.7 | 91.2 | 87.5 | 82.8 | 95.7 | 66.3 | 94.9 | 81.1 | 63.5 | 74.5 | 82.6 |
| Transformer | 83.42 | 85.1 | 82.9 | 85.4 | 87.7 | 78.8 | 90.5 | 80.8 | 91.1 | 87.7 | 85.3 | 95.6 | 73.9 | 94.9 | 83.5 | 61.2 | 74.9 | 80.6 |
| OcCo | 83.42 | 85.1 | 83.3 | 85.2 | 88.3 | 79.9 | 90.7 | 74.1 | 91.9 | 87.6 | 84.7 | 95.4 | 75.5 | 94.4 | 84.1 | 63.1 | 75.7 | 80.8 |
| Point-BERT | 84.11 | 85.6 | 84.3 | 84.8 | 88.0 | 79.8 | 91.0 | 81.7 | 91.6 | 87.9 | 85.2 | 95.6 | 75.6 | 94.7 | 84.3 | 63.4 | 76.3 | 81.5 |
| Point-MAE | 84.19 | 86.1 | 84.3 | 85.0 | 88.3 | 80.5 | 91.3 | 78.5 | 92.1 | 87.4 | 86.1 | 96.1 | 75.2 | 94.6 | 84.7 | 63.5 | 77.1 | 82.4 |
| ACT (Ours) | 84.66 | 86.14 | 85.2 | 85.2 | 88.8 | 81.2 | 91.3 | 79.4 | 92.2 | 87.9 | 85.8 | 96.0 | 75.5 | 95.5 | 85.2 | 66.6 | 77.7 | 81.5 |

## D  VISUALIZATION

**Reconstruction Results** Figure 3 compares the reconstruction results from our 2D image Transformer based 3D dVAE and Point-BERT 3D dVAE model. The results show that our 3D autoencoder can reconstruct high-quality details of the objects. For some relatively simple objects like the rectangular table in the second row, both our method and Point-BERT can reconstruct them well. However, for point sets with relatively complicated details, such as the thin shelf and armchair in the third row, our method can still reconstruct the object with detailed local geometric information. These qualitative observations are consistent with quantitative results in Table 7.

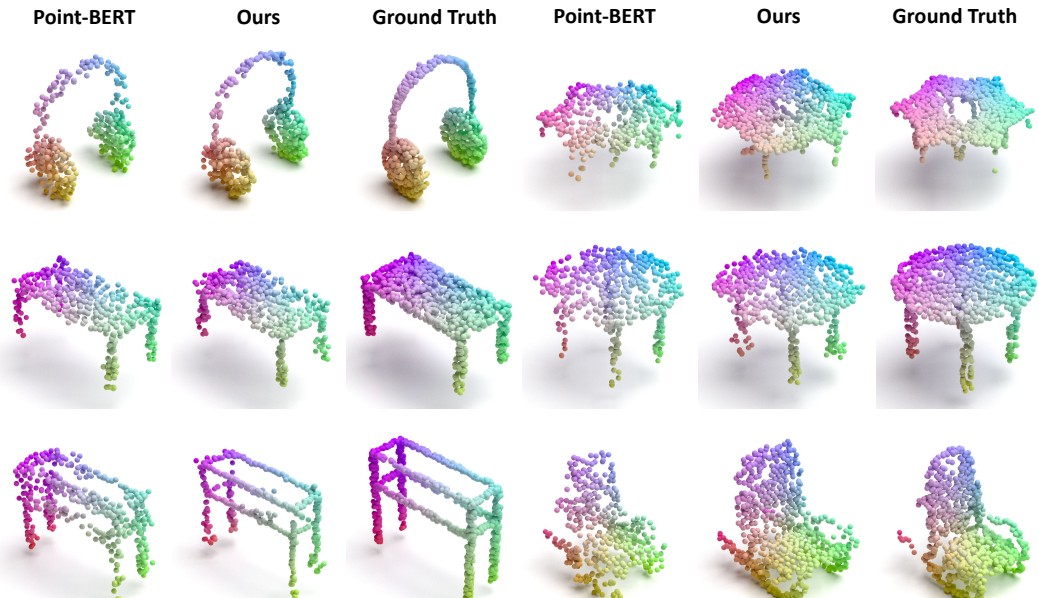

Figure 3:  Reconstruction results of synthetic objects from ShapeNet test set.

**t-SNE** Figure 4 shows the t-SNE (Van der Maaten & Hinton, 2008; Poličar et al., 2019) feature manifold visualization of models after pretraining on ShapeNet and fine-tuning on the ModelNet40 and ScanObjectNN PB_T50_RS dataset. It is observed that: (i) After pretraining on ShapeNet, the model can already yield discriminative features on ModelNet due to a relatively minor domain gap. (ii) After fine-tuning the downstream datasets, discriminative features are obtained on both ModelNet40 and the challenging ScanObjectNN datasets. (iii) The feature distribution extracted by ShapeNet-pretrained ACT on ScanObjectNN looks less discriminative. We argue that two reasons cause it: (i) the large domain gap between the synthetic ShapeNet and real-world ScanObjectNN datasets, and (ii) no contrastive loss for instance discrimination (*e.g.*, MoCo (He et al., 2020) loss used by Point-BERT (Yu et al., 2022)) is used by ACT. Interestingly, this yields better generalization performance on ScanObjectNN (88.21% OA of ACT versus 83.07% of Point-BERT).

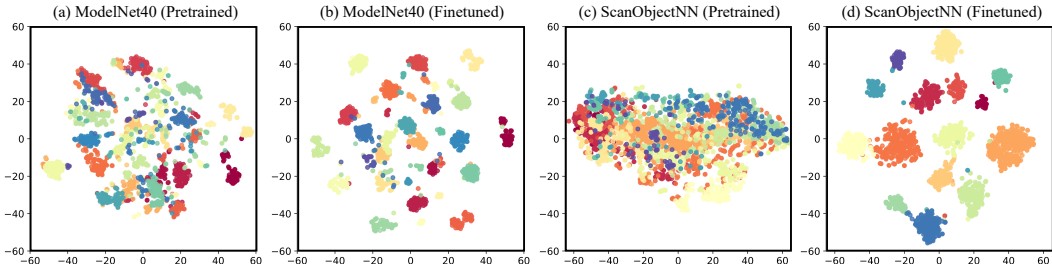

Figure 4:  t-SNE (Van der Maaten & Hinton, 2008) feature manifold visualization on ModelNet40 and ScanObjectNN PB_T50_RS datasets. Feature vectors extracted by ACT models after ShapeNet pretraining and downstream fine-tuning are visualized in (a), (c), and (b), (d), respectively.

