# OpenReview forum: "Autoencoders as Cross-Modal Teachers: Can Pretrained 2D Image Transformers Help 3D Representation Learning?"
_ICLR.cc/2023/Conference — ICLR 2023 poster_

### Official Review · Reviewer_8fVd · 2022-10-23

**Confidence:** 4
**Correctness:** 4
**Technical Novelty And Significance:** 4
**Empirical Novelty And Significance:** 4
**Recommendation:** 8

**Clarity, Quality, Novelty And Reproducibility:**

well-written

novel

see strengths and weaknesses.



**Strength And Weaknesses:**

== strength ==

- this paper is well-written and easy to follow. this paper is organized clearly. code is also provided for reproductivity.

- the experimental results on different tasks and datasets are convincing, showing the effectivenesses of the proposed method.

- the motivation is reasonable. the analysis for model pretraining on 3d and 2d is insightful including architecture, data, and pattern. based on the observation, the devised training strategy is also reasonable and efficient.

=== weaknesses ===

- I wonder how the model after the first stage performs.

- feature visualization is expected.

**Summary Of The Paper:**

This paper proposes an interesting strategy for transferring the knowledge from a model pretrained on 2D images to the 3D point cloud data. In detail, the developed strategy consists of two stages, one fine-tuning the pretrained models with new modules designed for 3D data and one learning a new 3D model via distilling knowledge from the model obtained in the first stage. The method is evaluated on three datasets, including ModelNet40, ScanObjectNN, and S3DIS, covering classification and segmentation.

The main contribution of this paper is the novel knowledge transfer strategy across different modalities.

**Summary Of The Review:**

This paper is well-written, and the proposed method is novel.

---

> ### Author Response · Authors · 2022-11-15
> **Response to Reviewer 8fVd**
>
> ---
>
> ***We sincerely appreciate your detailed and thoughtful reviews and hope our response can address your concerns.***
>
> ---
>
> **W1: Performance of first stage model?**
>
> **A:** Thanks for the constructive suggestions on fine-tuning the first-stage teacher models! Following your instructions, we fine-tune the first-stage ViT-B dVAE model on the ScanObjectNN dataset. From the results, we can observe that though the teacher model is relatively heavier with more parameters than the student, the fine-tuning results are not on par with the *pretrained* student (but way better than the from-scratch student). We argue that it is because the teacher model is a dVAE model that encodes clean input point clouds into representations for *reconstruction*. It is semantic-enriched and good to be used as denoised autoencoding targets. However, this auto-encoding fashion does not bring representations better for downstream generalization when compared to DAE pretraining. Therefore, the 3D student, as a DAE pretrained model, performs better than the teacher.
>
> | Transfer Protocol | Freeze | #Trainable Params | OBJ_BG | OBJ_ONLY | PB_T50_RS |
> | ----------------- | :----: | :---------------: | :----: | :------: | :-------: |
> | Full              |   ✗    |      114.42M      | 87.61  |  90.02   |   83.66   |
> | Full              |   ✓    |      29.36M       | 89.50  |  89.67   |   84.46   |
> | MLP-3             |   ✓    |       0.17M       | 80.38  |  82.44   |   68.29   |
> | MLP-Linear        |   ✓    |       0.01M       | 60.93  |  67.81   |   54.44   |
>
> ---
>
> **W2: Feature visualization is expected.**
>
> **A:** Thanks for the constructive suggestions about feature visualization! We have added feature manifold visualization with t-SNE [1, 2] and incorporated it into the revised paper. **Figure 4 in the Appendix** shows the visualization of models after pretraining on ShapeNet and fine-tuning on the ModelNet40 and ScanObjectNN PB_T50_RS dataset. It is observed that:
>
> - After pretraining on ShapeNet, the model can already yield discriminative features on ModelNet due to a relatively minor domain gap.
> - After fine-tuning the downstream datasets, discriminative features are obtained on both ModelNet40 and the challenging ScanObjectNN datasets.
> - The feature distribution extracted by ShapeNet-pretrained ACT on ScanObjectNN looks less discriminative. We argue that it is caused by two reasons: (i) the large domain gap between the synthetic ShapeNet and real-world ScanObjectNN datasets, and (ii) no contrastive loss for instance discrimination (*e.g.*, MoCo [3] loss used by Point-BERT [4]) is used here. Interestingly, this yields better generalization performance on ScanObjectNN (88.21% OA of ACT versus 83.07% of Point-BERT).
>
> [1] Van der Maaten & Hinton, Visualizing Data using t-SNE. JMLR, 2008.
>
> [2] Poliˇcar *et al.*, opentsne: a modular python library for t-sne dimensionality reduction and embedding. bioRxiv, 2019. URL https://github.com/pavlin-policar/openTSNE.
>
> [3] He *et al.*, Momentum contrast for unsupervised visual representation learning. In CVPR, 2020.
>
> [4] Yu *et al.*, Point-BERT: Pre-Training 3D Point Cloud Transformers With Masked Point Modeling. In CVPR, 2022.
>
> ---

---

### Official Review · Reviewer_FxNn · 2022-10-23

**Confidence:** 5
**Correctness:** 3
**Technical Novelty And Significance:** 3
**Empirical Novelty And Significance:** 3
**Recommendation:** 6

**Clarity, Quality, Novelty And Reproducibility:**

- The paper is clearly written and easy to follow in general. The best experimental results in ablation studies are expected to be highlighted to make it clearer.
- The whole pipeline seems like a combination of VPT (Visual Prompt Tuning), Point-BERT, and Point-MAE. Yet the idea of combining existing works to make use of multi-modal knowledge in 3D is interesting.
- Codes are provided.

**Strength And Weaknesses:**

# Strength:
- The paper leverages the knowledge from the image/language domains to improve the performance in 3D tasks, mitigating the lack of data in 3D.
- Instead of directly using 2D networks in 3D tasks, the paper constructs a two-step process to distill knowledge from 2D networks to 3D.
- Clear performance gain is observed in ScanObjectNN. Different settings (Full, MLP-Linear, MLP-3) are tested.

# Weaknesses:
- The performance on scene dataset S3DIS is far from state-of-the-art, though showing improvement over the baseline model. Is it possible to replace the backbone network with some stronger 3D networks to see if there is any improvement?
- It is not clear what causes the performance diversity with different teacher choices. Why do the vision models and language models perform better than language-vision models?
- In table 7, the DeiT-B dVAE without prompt has already achieved better performance than Point-BERT, especially the 'Freeze' one. In this case, what bridges the gap between 2D and 3D modalities? Is it related to the pre and post embedding network? Also, is it possible to change the g_pre and g_post to some more complicated structures to improve the performance?



**Summary Of The Paper:**

The paper proposes to leverage 2D pre-trained transformer models to help 3D understanding. The whole process follows two steps:
- The first is constructing a teacher model using the 2D / language knowledge via self-supervised prompt tuning.
- The second is pre-training a student 3D network via masked point modeling with the features from the teacher model as the guidance.



**Summary Of The Review:**

In general, I think this is an interesting paper to combine multi-modal knowledge. Some questions are listed in the weakness part.

---

> ### Author Response · Authors · 2022-11-15
> **Response to Reviewer FxNn (Part #1)**
>
> ---
>
> ***We sincerely appreciate your detailed and thoughtful reviews and hope our response can address your concerns.***
>
> ---
>
> **W1: Is it possible to replace the backbone network with some stronger 3D networks to see if there is any improvement?**
>
> **A:** Thanks for the valuable suggestions! The reasons why the S3DIS results are not on par with the state-of-the-art are three-fold:
>
> - **The current state-of-the-art methods use point cloud coordinates plus RGB colors as the input on S3DIS.** In comparison, our ACT is proposed for pure 3D geometric representation learning, which consumes only point cloud coordinates. We guess that the performance gap may come from this information utilization difference that RGB colors play a significant role in S3DIS segmentation.
> - **The pretraining dataset ShapeNet does not have color information for pretraining.** Besides, the pretraining dataset ShapeNet does not provide per-point color information like S3DIS, and thus it can not be directly applied here.
> - **ACT targets standard Transformer blocks, and it is non-trivial to use a stronger 3D architecture with such blocks.** This paper is targeted at 3D point cloud understanding with *standard* Transformer blocks, which are commonly more general and applicable than architectures with complicated computing blocks. However, this plain-style Transformer may need more inductive bias and local geometry understanding for the semantic segmentation task to offer satisfactory results. To design and incorporate architectures into standard Transformers with such features is beyond this work.
>
> Still, our ACT brings a promising improvement over the baseline model. Compared to the recently proposed 3D Transformer PCT [1], an advanced hierarchical and carefully-designed architecture consuming both point clouds and colors as input, ACT yields comparable or better results on S3DIS.
>
> **Further validation on 3D scene object detection.** Further, to show our ACT's effectiveness on state-of-the-art architectures with the scene-level understanding task. We conduct 3D object detection experiments on ScanNetV2 [2] with 3DETR [3], a recently proposed advanced plain Transformer 3D object detector. Following MaskPoint [4], we pretrain ACT on the subset of ScanNet with a sampling rate of 100. Due to limited time, We pretrain ACT for only 150 epochs, which is *half* of the training for MaskPoint. Still, we can achieve better performance than MaskPoint. The \#Epoch denotes the number of pretraining epochs.
>
> | Method            | \#Epochs | Detector | SSL  | Input | $\text{AP}_{25}$ | $\text{AP}_{50}$ |
> | ----------------- | -------- | :------: | :--: | :---: | :--------------: | :--------------: |
> | VoteNet [5]       | -        | VoteNet  |  ✗   | $xyz$ |       58.6       |       33.5       |
> | PointContrast [6] | -        | VoteNet  |  ✓   | $xyz$ |       59.2       |       38.0       |
> | STRL [7]          | -        | VoteNet  |  ✓   | $xyz$ |       59.5       |       38.4       |
> | RandomRooms [8]   | -        | VoteNet  |  ✓   | $xyz$ |       61.3       |       36.2       |
> | DepthContrast [9] | -        | VoteNet  |  ✓   | $xyz$ |       61.3       |        -         |
> | 3DETR [3]         | -        |  3DETR   |  ✗   | $xyz$ |       62.1       |       37.9       |
> | Point-BERT [10]   | 300      |  3DETR   |  ✓   | $xyz$ |       61.0       |       38.3       |
> | MaskPoint [4]     | 300      |  3DETR   |  ✓   | $xyz$ |       63.4       |       40.6       |
> | ACT (Ours)        | **150**  |  3DETR   |  ✓   | $xyz$ |     **63.5**     |     **41.0**     |
>
> [1] Guo *et al.*, PCT: Point Cloud Transformer. Comput. Vis. Media, 2021.
>
> [2] Dai *et al.*, Scannet: Richly-Annotated 3D Reconstructions of Indoor Scenes. In CVPR, 2017.
>
> [3] Misra, Girdhar & Joslin, An End-to-End Transformer Model for 3D Object Detection. In ICCV, 2021.
>
> [4] Liu, Cai & Lee, Masked Discrimination for Self-Supervised Learning On Point Clouds. In ECCV, 2022.
>
> [5] Qi *et al.*, Deep Hough Voting for 3D Object Detection in Point Clouds. In ICCV, 2019.
>
> [6] Xie *et al.*, PointContrast: Unsupervised Pretraining for 3D Point Cloud Understanding. In ECCV, 2020.
>
> [7] Huang *et al.*, Spatio-temporal Self-Supervised Representation Learning for 3D Point Clouds. In ICCV, 2021.
>
> [8] Rao *et al.*, RandomRooms: Unsupervised Pretraining from Synthetic Shapes and Randomized Layouts for 3D Object Detection. In ICCV, 2021.
>
> [9] Zhang *et al.*, Self-Supervised Pretraining of 3D Features on any Point-Cloud. In ICCV, 2021.
>
> ---

---

> > ### Author Response · Authors · 2022-11-15
> > **Response to Reviewer FxNn (Part #2)**
> >
> > ---
> >
> > **W2: Why do vision and language models perform better than language-vision models?**
> >
> > **A:** Thanks for pointing out this interesting question! We answer this question from the following perspectives:
> >
> > * **The modality gap between the cross-modal teacher and the single-modal student may be more significant.** The primary difference between these single-modal models (**i.e.**, vision or language models) and the cross-modal models (**i.e.**, language-vision models) is that the single-modal models are trained with either human-annotated supervision or semantically rich language self-supervision, while the language-vision models (*i.e.*, CLIP [5]) are self-supervised and trained to align between languages and images. We hypothesize that this strong alignment between images and texts is able to yield semantic-rich features between images and languages, but it may limit the representational adaption for modalities other than images or texts (like point clouds used here). Therefore, as the 3D autoencoding is a single-modality representation adaptation, the modality gap between the cross-modal teacher and the single-modal student may be more significant.
> > * **CLIP focuses more on global representation, which may not be optimal for cross-modal dense masked modeling.** In comparison to vision-only pretrained ViTs, CLIP is known to have a biased representation of global semantics [11], which may not be suitable for dense tasks such as semantic segmentation, as pointed out by Li *et al.* [12]. Masked point modeling is a dense pretraining task, which may rely on more local geometric feature encoding for the semantic-sparse point cloud inputs. Therefore, CLIP may not be optimal here for cross-modal dense masked modeling.
> > * **ACT brings promising results for such cross-modal language-vision teachers.** Still, with our proposed ACT, the CLIP-B teacher can consume and encode 3D point clouds and facilitate 3D representation learning. This exciting observation is worth future efforts for the fundamental reasoning understanding, which remains open and could spur more related works in the cross-modal learning community.
> >
> > [10] Radford *et al.*, Learning Transferable Visual Models From Natural Language Supervision. In ICML, 2021.
> >
> > [11] Goh *et al.*, Multimodal Neurons in Artificial Neural Networks. Distill 6, no. 3 (2021): e30. URL https://distill.pub/2021/multimodal-neurons/?utm_campaign=Dynamically%20Typed&utm_medium=email&utm_source=Revue%20newsletter
> >
> > [12] Li *et al.*, Language-driven Semantic Segmentation. In ICLR 2022.
> >
> > ---
> >
> > **W3: What bridges the gap between 2D and 3D modalities regarding results in Table 7?**
> >
> > **A:** Thanks for pointing out this insightful and constructive question! We first clarify the results in Table 7, then offer an experimental analysis following your instructions, and finally provide a discussion for answering the question.
> >
> > **Clarification on the dVAE trainable parameters.** Please note that for the 'Freeze' dVAE teacher in Table 7, the trainable parameters include not only the embedding networks $g^{\text{pre}}$ and $g^{\text{post}}$, but also the prompt tokens (optional) and a FoldingNet decoder [13]. Under the end-to-end reconstruction self-supervision, $g^{\text{pre}}$ learns to embed point cloud inputs into tokens that the pretrained 2D Transformer can consume. In contrast, the $g^{\text{post}}$ learns to embed the encoded latent feature for the FoldingNet decoding. Besides, this DGCNN-style embedding network follows the original architectural design of FoldingNet.
> >
> > [13] Yang *et al.*, Foldingnet: Point Cloud Auto-encoder via Deep Grid Deformation. In CVPR, 2018.
> >
> > **Experimental analysis on more complicated embedding networks.** Thanks for your suggestions on using more complicated embedding networks! We have changed the $g^{\text{pre}}$ and $g^{\text{post}}$ embedding networks from simple DGCNN block into PointMLP [14] block, which is recently proposed as a modern architecture for 3D point cloud understanding. The first stage ViT-B dVAE training results are shown in the following table, where the #Params denotes the number of parameters of the embedding network:
> >
> > | Embedding Block | #Params | Num. of Prompt | Prompt Type | Freeze | F-Score $\uparrow$ | CD-$\ell_1 \downarrow$ | CD-$\ell_2 \downarrow$ |
> > | --------------- | ------- | :------------: | :---------: | :----: | :----------------: | :--------------------: | :--------------------: |
> > | DGCNN           | 23.4M   |       64       |    deep     |   ✓    |       0.193        |         23.524         |         2.110          |
> > | PointMLP [14]    | 87.8M   |       64       |    deep     |   ✓    |       0.155        |         26.334         |         2.933          |
> >
> > [14] Xu et al., Rethinking Network Design and Local Geometry in Point Cloud: A Simple Residual MLP Framework. In ICLR, 2022.

---

> > > ### Author Response · Authors · 2022-11-15
> > > **Response to Reviewer FxNn (Part #3)**
> > >
> > > **Complicated embedding networks may not help modality gap reduction.** From the above results, we observe that the modified dVAE model with a heavier and more complicated embedding network leads to inferior performance to the lightweight DGCNN embedding network. We argue it is because:
> > >
> > > * The DGCNN-style FoldingNet is a carefully-designed dedicated dVAE architecture that requires non-trivial crafting efforts and directly modifying it generally leads to unsatisfactory results.
> > >
> > > * The embedding network is used to reduce the modality gap by providing *low-semantic inputs* for the pretrained Transformer understanding. On the contrary, a complicated embedding network may provide a *3D-biased* representation with high-level semantics that enlarges the modality gap. This representation is hard to be understood by the pretrained cross-modal Transformers, thus poor representations are obtained.
> > >
> > > This is also closely related to the intrinsic feature of Transformers. We discuss this as follows:
> > >
> > > **Transformers can be used to provide modality-agnostic semantic representation.** Another important factor that bridges the modality gap may come from the Transformer's inherent architectural feature. The Transformer is built with stacked multi-head self-attention (MSA). It is known to have a strong representational capacity to model both long- and short-range dependences of the input sequential embeddings, encoding high-level semantics with early aggregation [15]. Besides, Transformers are demonstrated to have a capability of *in-context* learning [16], *e.g.*, the representation of a pretrained Transformer can be used to learn unseen functions without parameter updating [17]. This motivates ACT to encode cross-modal inputs into low-semantic embeddings to fit the pretrained Transformer for modality gap reduction. Therefore, strong and unified representation can thus be output by the pretrained foundational Transformer with high-level semantics. We hypothesize that: this *semantic-enriched representation with a reduced modality gap* can be seen as *modality-agnostic*, which may be a critical factor that bridges the modality gap.
> > >
> > > [15] Raghu *et al.*, Do Vision Transformers See Like Convolutional Neural Networks? In NeurIPS 2021.
> > >
> > > [16] Brown *et al.*, Language Models are Few-Shot Learners. In NeurIPS 2020.
> > >
> > > [17] Garg *et al.*, What Can Transformers Learn In-Context? A Case Study of Simple Function Classes. arXiv preprint arXiv:2208.01066 (2022).
> > >
> > > **Positional embeddings also play a significant role in modality gap reduction.** As shown in Table 10, the *positional embeddings* also play a significant role in the modality gap reduction. Here we make a discussion about the analysis in Table 10:
> > >
> > > * **Modality gap hypothesis.** 2D images are regularly distributed on grids with RGB pixels as basic units, while 3D point clouds are irregularly sampled in 3D Cartesian coordinate space. We ask: *are the 3D point cloud inputs encoded (or projected) as some plane in 2D space that the 2D Transformers can understand?* In this case, we want to determine if the **modality gap** comes from the **data pattern difference** between 2D images and 3D point clouds. To this end, the following three positional encoding settings are conducted.
> > > * **Table 10 (a)** shows that *without* explicit positional embeddings, the pretrained 2D Transformer can hardly understand the 3D point cloud inputs. However, due to the solid representational capacity, it still brings promising transfer learning performance. We argue that this is because of the unique feature of 3D point clouds that its inputs already contains its positional information.
> > > * **Table 10 (b)** shows that *with the 2D $xy$ plane coordinate* as positional embeddings, the 2D Transformers succeed in understanding 3D inputs, and significant performance improvements have been observed. This demonstrates that 3D inputs with 2D positional embedding are able to reduce the modality gap largely, and better feature representation is utilized.
> > > * **Table 10 (c)** shows that *with all 3D point clouds $xyz$* as positional embeddings, the 2D Transformers can further leverage the additional information beyond 2D plane space. It demonstrates the strong representational capacity of the pretrained 2D Transformers, and the modality gap is thus significantly reduced.
> > >
> > > In summary, the ablation study in Table 10 shows that different positional encodings help bridge the modality gap between 2D and 3D, making the 2D models understand 3D data. Besides, from Table 7, we see that the frozen 2D Transformer succeeds in improving the understanding of 3D inputs with the use of cross-modal prompts. All these factors enable the possibility of using the modality-agnostic representations output by the pretrained foundational Transformers.
> > >
> > > ---

---

### Official Review · Reviewer_8SEp · 2022-10-24

**Confidence:** 4
**Correctness:** 4
**Technical Novelty And Significance:** 4
**Empirical Novelty And Significance:** 4
**Recommendation:** 10

**Clarity, Quality, Novelty And Reproducibility:**

### Clarity
* Again, the paper is well-written. Adequate references to earlier contributions are also provided.
### Novelty
* As mentioned in the strength section, the proposed approach is technically sound and novel.
### Reproducibility
* The code has been included in the supplemental material with clear documentation, which should be sufficient to reproduce the experiments.



**Strength And Weaknesses:**

### Strength
* The paper is well-written and very enjoyable to read.
* The proposed approach is novel and technically sound. This is the first work showing that a pre-trained 2D vision transformer can
help 3D representation learning without accessing any 2D data.
* The experiment setups and analysis are comprehensive. Moreover, the paper offers informative insights with several interesting discussions.
* The improvements are significant. The proposed framework ACT consistently performs well against the other baselines by a large margin.
### Weaknesses
I found no major weaknesses.

**Summary Of The Paper:**

The paper proposes a cross-modal feature distillation framework for 3D representation learning. Specifically, the proposed framework performs masked modeling as feature distillation from pre-trained 2D image Transformer to 3D Transformer students. Experiments show that the proposed framework has satisfactory generalization performance.

**Summary Of The Review:**

This paper may inspire future work in transferring cross-modal knowledge in a self-supervised fashion.

---

> ### Author Response · Authors · 2022-11-15
> **Response to Reviewer 8SEp**
>
> We sincerely thank you for the solid positive review, and we are glad to hear that you are pleased with our paper! ACT is a general self-supervised cross-modal representation learning framework, and we hope this ACT-style representation learning could spur more related works in the future.

---

### Official Review · Reviewer_Dx8R · 2022-10-27

**Confidence:** 5
**Correctness:** 2
**Technical Novelty And Significance:** 3
**Empirical Novelty And Significance:** 2
**Recommendation:** 5

**Clarity, Quality, Novelty And Reproducibility:**

The paper is well-written with some novelty concerning the construction of cross-modal teachers, but fail to clarity its main contribution: why 2D helps 3D?

**Strength And Weaknesses:**

Strength:
1) The problem ACT aims to solve is practical and reasonable: insufficient 3D data can not afford large-scale pre-training like 2D and languages. The motivation is also clearly illustrated in the introduction that pre-trained models from other modalities with cross-modal teaching might be helpful.
2) Writing and figures are presented with high qualities. It is easy to follow and understand the proposed multi-step method.
3) The idea to construct cross-modal teachers by intermediate antoencoding and prompting is interesting.

Weakness:
1) The first concern is the too much complicated pipeline. As summarized above, for a downstream task, the pre-training requires two steps of pre-training (actually three steps if 2D's is included), which is time/space-consuming. The reviewer is suspicious about utilization values of ACT. The author could give some pre-training efficiency comparison with existing 3D self-supervised methods.
2) It is unclear whether the 2D models indeed helps 3D understanding. Although the authors try to explain it in the discussion section, it still remains black-box.

>**(a)** As shown in Figure 2 (b), using BERT-B pre-trained by languages as the teacher seems to surpass all of the vision models except ViT-B. If this results indicate that a good pre-trained transformer as initialization is already working for ACT, no matter it is a 2D model or not?

>**(b)** In Table 8, the paper only utilizes Point-BERT to replace 2D teachers. Considering Point-BERT itself underperforms the newer PointMLP, MaskPoint and Point-MAE, using Point-BERT for ablation is not convincing to verify the effectiveness of ACT's 2D teachers. What happens if using Point-MAE here? Will the simple 3D teacher be better than 2D?

>**(c)** The reviewer fails to understand the analysis about Table 10, e.g. why different positional encodings reveal 2D model understands 3D?

3) The performance of ACT is not strong enough.

>**(a)** Why ACT performs a little worse than Point-MAE on ModelNet40 (93.7 vs 93.8)? Given that ACT completely inherits network architectures of Point-MAE, if this means the simple 3D MIM in Point-MAE is already better than ACT's complicated two-step pre-training on synthetic data?

>**(b)** On ShapeNetPart, ACT can hardly bring enhancement over Point-MAE (86.14 vs 86.1).

**Summary Of The Paper:**

1) The paper proposes ACT to utilize 2D pre-trained models as teachers for cross-modality knowledge transfer by masked point modeling. This can alleviate the data desert problem in 3D by using rich semantics learned from 2D images.
2) ACT consists of three steps: 1. Adapt the pre-trained 2D/language models into 3D teachers via DAE and prompting on ShapeNet. 2. Regard this transferred model as distillation teachers to pre-train a Point-MAE. 3. Fine-tune the pre-trained Point-MAE on specific downstream tasks, e.g., classification, segmentation.
3) ACT achieves favorable performance on various downstream benchmarks. The related discussion solves part of the confusions.

**Summary Of The Review:**

The reviewer expects authors to respond to above questions in the weakness, which largely matter the contribution of this paper.

---

> ### Author Response · Authors · 2022-11-15
> **Response to Reviewer Dx8R (Part #1)**
>
> ---
>
> ***We sincerely appreciate your detailed and thoughtful reviews and hope our response can address your concerns.***
>
> ---
>
> **Clarifications on Possible Misunderstandings** We sincerely appreciate your detailed and constructive reviews! From your summary of our paper, we find that there might exist some possible misunderstandings. We are sorry for the confusion, and we want first to make the following clarifications to avoid unnecessary misunderstandings:
>
> - **Clarification on stage I teacher training.** In the first stage of ACT, the pre-trained 2D/language Transformers are adpated into 3D teachers via *discrete Variational Auto Encoding* (**dVAE** [1, 2]) but **not** *Denoising Auto Encoding* (**DAE** [3, 4]). dVAE and DAE are *Auto Encoders* (**AE**), but their differences are apparent. dVAE is a class of variational autoencoders that reconstructs by using discrete distributions (*e.g.*, uniform distribution) as prior of the latent variables. In contrast, DAE is a class of autoencoders that first corrupt the input signal and then reconstruct it.
>
> - **Clarification on stage II student training.** We want to clarify that our ACT student is not a Point-MAE [6], and their differences are clear. The following table lists a comprehensive comparison of Point-MAE and our ACT. The notations are taken from Sec. 3.2.
>
>   | Method        |               Teacher $f_{\mathcal{T}}$                | Student $f_{\mathcal S}$ | Masked Modeling Head $h$ | Modeling Metric $\mathcal L_{\mathbb D}$  |   Modeling Target   |
>   | ------------- | :----------------------------------------------------: | :----------------------: | :----------------------: | :---------------------------------------: | :-----------------: |
>   | Point-MAE [6] |                    Identity Mapping                    |   Vanilla Transformer    |            FC            |       Chamfer Distance CD-$\ell_2$        | 3D Coordinate $xyz$ |
>   | ACT (Ours)    | 3D Autoencoders w/ pretrained 2D/language Transformers |   Vanilla Transformer    |            FC            | Cosine Distance, $\ell_1$, $\ell_2$, etc. |       Feature       |
>
> Overall, Point-MAE can be viewed as the 3D variant of 2D MAE [5], which has **no** *parameterized teacher*, and the teacher is an *identity mapping*. (Please also see our preliminary discussion in Sec. 3.2 to have a unified understanding of masked modeling from a knowledge distillation perspective.) Besides, the *masked reconstruction target* is the input 3D coordinates, and the *metric function* is Chamfer Distance, explicitly designed for 3D point cloud reconstruction. In terms of the characters in common, these methods can all be viewed as an extended version of DAE-style representation learning, as pointed out by He *et al.* [5].
>
> [1] Ramesh *et al.*, Zero-Shot Text-to-Image Generation. In ICML, 2022.
>
> [2] Yu *et al.*, Point-BERT: Pre-Training 3D Point Cloud Transformers With Masked Point Modeling. In CVPR, 2022.
>
> [3] Vincent *et al.*, Extracting and Composing Robust Features with Denoising Autoencoders. In ICML, 2008.
>
> [4] Vincent *et al.*, Stacked Denoising Autoencoders: Learning Useful Representations in a Deep Network with a Local Denoising Criterion. JMLR, 2010.
>
> [5] He *et al.*, Masked Autoencoders Are Scalable Vision Learners. In CVPR, 2022.
>
> [6] Pang *et al.*, Masked Autoencoders for Point Cloud Self-supervised Learning. In ECCV, 2022.
>
> ---
>
> **W1: Pretraining efficiency comparison.**
>
> **A:** Thanks for your suggestions on providing the pretraining efficiency comparison! In the following table, we compare the pretraining GPU hours on a single Titan 2080Ti GPU device of our ACT to **(a)** Point-BERT, **(b)** Point-MAE, **(c)** Point-MAE with 4$\times$pretraining configuration, and **(d)** Point-MAE with our ACT as auxiliary knowledge distillation (Table 9 in the paper). Note that we did not use distributed training and all the batch size are the same except for Point-BERT, which is reduced by 12% due to limited memory budgets on 2080Ti.
>
> |  Id  | Method                             | Stage I | Stage II | Total |     OA (%)     |
> | :--: | ---------------------------------- | :-----: | :------: | :---: | :------------: |
> | (a)  | Point-BERT [2]                     |   25h   |   40h    |  65h  |     83.07      |
> | (b)  | Point-MAE [6]                      |   N/A   |   18h    |  18h  | 84.35$\pm$0.31 |
> | (c)  | Point-MAE [6] + 4$\times$ training |   N/A   |   72h    |  72h  | 84.49$\pm$0.49 |
> | (d)  | Point-MAE [6] + Ours KD            |   36h   |   38h    |  74h  | 84.96$\pm$0.58 |
> | (e)  | ACT (Ours)                         |   36h   |   36h    |  72h  | 85.33$\pm$0.27 |

---

> > ### Author Response · Authors · 2022-11-15
> > **Response to Reviewer Dx8R (Part #2)**
> >
> > It is observed:
> >
> > - ACT costs 1.1$\times$ longer pretraining time than **(a)** Point-BERT with +2.26% overall accuracy and 4$\times$ longer pretraining time than **(b)** Point-MAE with +0.98% overall accuracy;
> > - When **(c)** prolonging the pretraining time of Point-MAE to the same as ACT, the performance gap between ACT is shorted but still huge (-0.84% overall accuracy gap);
> > - When **(d)** using ACT as auxiliary knowledge distillation to facilitate Point-MAE pretraining, the time cost is comparable to ACT, and the performance gap is reduced to -0.37% overall accuracy.
> > - Our **(e)** ACT shows superior generalization performance with data-efficient pretraining, and the pretraining cost is acceptable and applicable.
> >
> > ---
> >
> > **W2: Unclear whether 2D models indeed help 3D understanding.**
> >
> > **W2.1: Is a good pre-trained transformer as initialization already working for ACT, no matter whether it is a 2D model or not?**
> >
> > **A:** **Yes,** the preatreined foundational Transformer is not necessarily a 2D model.  Indeed, our ACT:
> >
> > - aims at utilizing *foundational Transformers* pretrained on *data larger than 3D point clouds* to facilitate *self-supervised 3D representation learning*, which is not limited to 2D models,
> > - and it is a *general* framework that can be easily extended to other modalities.
> >
> > In this paper, we mainly use the pretrained 2D image Transformers as the masked modeling teacher to validate our method. This foundational Transformer can also be a language model pretrained on semantic-rich natural languages, *e.g.*, BERT$_{\text{base}}$ as shown in Figure 2 (b). This shows that our method can generalize to other modalities like languages. We have revised the paper to avoid this confusion.
> >
> > ---
> >
> > **W2.2: What happens if using Point-MAE to verify the effectiveness of the 2D teacher?**
> >
> > **A:** Thanks for your valuable suggestions! Here we make a clarification on the results in Table 8.
> >
> > **Clarification on Table 8 Results**
> >
> > - **The teacher we use is a simple 3D teacher.** Firstly, we want to clarify that the "Point-BERT" in Table 8 *Teacher* column represents the **dVAE teacher of Point-BERT** but **not** the Point-BERT student, which is indeed a *simple 3D autoencoder teacher*.
> > - **Point-BERT targets can be directly applied for a fair comparison.** The "Point-BERT" in Table 8 *Target* column represents the **discrete token classification** target, which can be applied here for a fair comparison since our ACT teacher is also a 3D dVAE model that can encode 3D inputs into discrete representations.
> > - **Point-MAE has no parameterized teacher.** As shown in the above table in our *clarification on stage II student training*, Point-MAE does **not** have a **parametrized teacher** like Point-BERT or ACT. Therefore, we did not use Point-MAE to validate the effectiveness of 2D Transformers of the teacher.
> >
> > ---
> >
> > **W2.3: Why do different positional encodings reveal 2D model understands 3D?**
> >
> > **A:** Sorry for the confusion. In the following, we make a detailed clarification of this ablation study in Table 10.
> >
> > - **Modality gap hypothesis.** 2D images are regularly distributed on grids with RGB pixels as basic units, while 3D point clouds are irregularly sampled in 3D Cartesian coordinate space. We ask: *are the 3D point cloud inputs encoded (or projected) as some plane in 2D space that the 2D Transformers can understand?* In this case, we want to determine if the **modality gap** comes from the **data pattern difference** between 2D images and 3D point clouds. To this end, the following three positional encoding settings are conducted.
> > - **Settings.** In Table 10, we have conducted an ablation study on the effect of positional encodings with three different settings, *i.e.*, **(a) N/A**: no positional embedding is used, **(b) 2D/$z$**: positional embedding with only 2D $xy$ Plane coordinates, and **(c) 3D**: positional embedding with all 3D $xyz$ coordinates.
> > - **Table 10 (a)** shows that *without* explicit positional embeddings, the pretrained 2D Transformer can hardly understand the 3D point cloud inputs. However, due to the solid representational capacity, it still brings promising transfer learning performance. We argue that this is because of the unique feature of 3D point clouds that its inputs already contains its positional information.
> > - **Table 10 (b)** shows that *with the 2D $xy$ plane coordinate* as positional embeddings, the 2D Transformers succeed in understanding 3D inputs, and significant performance improvements have been observed. This demonstrates that 3D inputs with 2D positional embedding are able to largely reduce the modality gap, and better feature representation is utilized.
> > - **Table 10 (c)** shows that *with all 3D point clouds $xyz$* as positional embeddings, the 2D Transformers can further leverage the additional information beyond 2D plane space. It demonstrates the strong representational capacity of the pretrained 2D Transformers, and the modality gap is thus significantly reduced.

---

> > > ### Author Response · Authors · 2022-11-15
> > > **Response to Reviewer Dx8R (Part #3)**
> > >
> > > - **Conclusion.** The ablation study in Table 10 shows that different positional encodings help bridge the modality gap between 2D and 3D, making the 2D models understand 3D data.
> > >
> > > ---
> > >
> > > **W3: The performance of ACT is not strong enough.**
> > >
> > > **A:** In the following, we demonstrate that the performance of ACT is strong enough from the following perspectives:
> > >
> > > - **This 0.1% performance gap on ModelNet40 doesn't indicate ACT is not strong.** It is known that synthetic datasets like ModelNet40 and ShapeNetPart are relatively simpler, where the performance is already saturated and hard to distinguish. Still, our ACT has improved the from-scratch baseline by a large margin, and results comparable to the state-of-the-art models have been achieved. Therefore, these results show that for these synthetic datasets, our ACT and MAE-style 3D masked point modeling (MPM) are all working well. Besides, our ACT can also be utilized as *auxiliary knowledge distillation* to improve such 3D MPM learning to provide more data-efficient pretraining. For example, the overall accuracy of Point-MAE on ScanObjectNN is improved from 85.18% to 86.05% with our ACT as the auxiliary distillation.
> > >
> > > - **ACT yields strong performance on more challenging real-world datasets.** In contrast, real-world datasets like ScanObjectNN and S3DIS are much more challenging but with great practical value. Significant progress is being witnessed toward a better perception of these scenarios. On ScanObjectNN, our ACT achieves a state-of-the-art overall accuracy (*i.e.*, 88.21% on the most challenging PB_T50_RS benchmark). While the 3D MPM method is limited when applied to real-world datasets or scaled up with extended learning configurations. Further, with voting strategy [7], these results can be further boosted to 89.17%. This demonstrates the strong performance of ACT on these real-world datasets.
> > >
> > > - **Further validation on 3D scene object detection.** In order to further validate our ACT on more scene-level real-world datasets, we conduct 3D object detection experiments on ScanNetV2 [8] with 3DETR [9], a recently proposed advanced plain Transformer 3D object detector. Following MaskPoint [10], we pretrain ACT on the subset of ScanNet with a sampling rate of 100. Due to limited time, We pertrain ACT for only 150 epochs, which is *half* of the training for MaskPoint. Still, we can achieve better or comparable performance than MaskPoint. The \#Epoch denotes the number of pretraining epochs.
> > >
> > >
> > > | Method            | \#Epochs | Detector | SSL  | Input | $\text{AP}_{25}$ | $\text{AP}_{50}$ |
> > > | ----------------- | -------- | :------: | :--: | :---: | :--------------: | :--------------: |
> > > | VoteNet [5]       | -        | VoteNet  |  ✗   | $xyz$ |       58.6       |       33.5       |
> > > | PointContrast [6] | -        | VoteNet  |  ✓   | $xyz$ |       59.2       |       38.0       |
> > > | STRL [7]          | -        | VoteNet  |  ✓   | $xyz$ |       59.5       |       38.4       |
> > > | RandomRooms [8]   | -        | VoteNet  |  ✓   | $xyz$ |       61.3       |       36.2       |
> > > | DepthContrast [9] | -        | VoteNet  |  ✓   | $xyz$ |       61.3       |        -         |
> > > | 3DETR [3]         | -        |  3DETR   |  ✗   | $xyz$ |       62.1       |       37.9       |
> > > | Point-BERT [10]   | 300      |  3DETR   |  ✓   | $xyz$ |       61.0       |       38.3       |
> > > | MaskPoint [4]     | 300      |  3DETR   |  ✓   | $xyz$ |       63.4       |       40.6       |
> > > | ACT (Ours)        | **150**  |  3DETR   |  ✓   | $xyz$ |     **63.5**     |     **41.0**     |
> > >
> > > [7] Liu *et al.*,  Relation-Shape Convolutional Neural network for Point Cloud Analysis. In CVPR, 2019.
> > >
> > > [8] Dai *et al.*, Scannet: Richly-Annotated 3D Reconstructions of Indoor Scenes. In CVPR, 2017.
> > >
> > > [9] Misra, Girdhar & Joslin, An End-to-End Transformer Model for 3D Object Detection. In ICCV, 2021.
> > >
> > > [10] Liu, Cai & Lee, Masked Discrimination for Self-Supervised Learning on Point Clouds. In ECCV, 2022.
> > >
> > > [11] Qi *et al.*, Deep Hough Voting for 3D Object Detection in Point Clouds. In ICCV, 2019.
> > >
> > > [12] Xie *et al.*, PointContrast: Unsupervised Pretraining for 3D Point Cloud Understanding. In ECCV, 2020.
> > >
> > > [13] Huang *et al.*, Spatio-temporal Self-Supervised Representation Learning for 3D Point Clouds. In ICCV, 2021.
> > >
> > > [14] Rao *et al.*, RandomRooms: Unsupervised Pretraining from Synthetic Shapes and Randomized Layouts for 3D Object Detection. In ICCV, 2021.
> > >
> > > [15] Zhang *et al.*, Self-Supervised Pretraining of 3D Features on any Point-Cloud. In ICCV, 2021.
> > >
> > > ---

---

> > > > ### Author Response · Authors · 2022-11-15
> > > > **Response to Reviewer Dx8R (Part #4)**
> > > >
> > > > ---
> > > >
> > > > **Main Contribution Clarification** The main contribution of this paper is to offer a general framework ACT for self-supervised 3D representation learning, where a pretrained foundational Transformer **can** help the representation learning of 3D models without accessing any other modality data or 3D downstream annotations. In terms of the reason **why** the foundational Transformers could help 3D representation learning, we have made great efforts to find it out with experimental analysis in Sec. 5 and 6, and the reasons are concluded as follows:
> > > >
> > > > 1. The *proposed ACT* enables the foundational Transformers pretrained on 2D images or natural languages intrinsically *cross-modal teachers* that can consume and encode 3D inputs in a self-supervised fashion.
> > > > 2. Parameter-efficient prompt tuning facilitates the pretrained cross-modal Transformer autoencoding adaptation and the foundational knowledge inheritance, as shown in Table 7.
> > > > 3. Appropriate *targets with enriched semantics* facilitate masked DAE-style representation learning of unstructured and semantic-sparse  3D point clouds, as shown in Table 8.
> > > > 4. Appropriate *positional encoding* can vastly reduce the *data pattern difference* modality gap, as shown in Table 10.
> > > > 5. **Transformers can be used to provide modality-agnostic semantic representation.** Another important reason why ACT works may come from the Transformer's inherent architectural feature. The Transformer is built with stacked multi-head self-attention (MSA), and it is known to have a strong representational capacity to model both long- and short-range dependences of the input sequential embeddings, encoding high-level semantics with early aggregation [16]. Besides, Transformers are demonstrated to have a capability of *in-context* learning [17], *e.g.*, the representation of a pretrained Transformer can be used to learn unseen functions without parameter updating [18]. This motivates ACT to encode cross-modal inputs into low-semantic embeddings to fit the pretrained Transformer for modality gap reduction, while strong and unified representation can thus be output by the pretrained foundational Transformer with high-level semantics. We hypothesize that: this *semantic-enriched representation with a reduced modality gap* can be seen as *modality-agnostic*, which may be one important reason why ACT works.
> > > >
> > > > We are interested in studying the reason why ACT works, and it is our future work to offer a more fundamental understanding, like the theoretical analysis.
> > > >
> > > > [16] Raghu *et al.*, Do Vision Transformers See Like Convolutional Neural Networks? In NeurIPS 2021.
> > > >
> > > > [17] Brown *et al.*, Language Models are Few-Shot Learners. In NeurIPS 2020.
> > > >
> > > > [18] Garg *et al.*, What Can Transformers Learn In-Context? A Case Study of Simple Function Classes. arXiv preprint arXiv:2208.01066 (2022).
> > > >
> > > > ---

---

### Author Response · Authors · 2022-11-22
**Sincere Request for Further Discussions**

Dear Reviewers,

Thanks again for your great efforts in reviewing this paper! With the discussion period drawing to a close, we expect your feedback and thoughts on our reply. We put a significant effort into our response, with several new experiments and discussions. We sincerely hope you can consider our reply in your assessment. We look forward to hearing from you, and we can further address unclear explanations and remaining concerns if any.

Regards, \
Authors

---

### Decision · Program_Chairs · 2023-01-20

**Decision:**

Accept: poster

**Justification For Why Not Higher Score:**

While the paper received an overall very high score, two of the reviews raise very valid concerns regarding the quality of the experimental results and the complicated pipeline.

**Justification For Why Not Lower Score:**

The paper proposes an interesting setting and addresses an important problem. The paper is also very well presented. I enthusiastically recommend to accept this paper.

**Metareview: Summary, Strengths And Weaknesses:**

The paper proposes to use pretrained image transformers, autoencoders, as cross-modal teachers for 3D representation learning. The resulting approach is called ACT. It uses information from 2D pre-trained models to introduce cross-modality knowledge by masked point modeling. ACT can thereby address the data desert problem. The paper shows experimentally that ATC shows overall good performance in several relevvant setting.
The paper has received scores ranging from 5 to 10, where all reviewers agree that the proposed idea is novel and interesting, the addressed problem is highly relenvant since 3D data for training is usually scarse, and that the paper is very well written, presented and illustrated.
The main criticism is the performance of the resulting models, which is not good enough in all cases. In particual, the low performance on the scene dataset S3DIS is significantly below the state of the art. At the same time, the proposed framework is very complicated, making it hard to attribute certain improvements to specific model aspects.

**Note From Pc:**

if the above contains the word "oral" or "spotlight" please see: "oral" presentation means -> notable-top-5% and "spotlight" means -> notable-top-25%. As stated in our emails, we are disassociating presentation type from AC recommendations